



# Sources and sinks of carbonyl sulfide inferred from tower and mobile atmospheric observations

Alessandro Zanchetta[1], Linda Maria Johanna Kooijmans[1,3], Steven van Heuven[1], Andrea Scifo[1], Hubertus A. Scheeren[1], Ivan Mammarella[4], Ute Karstens[5], Jin Ma[6], Maarten Krol[3,6], Huilin Chen[1,2]

[1]*Centre for Isotope Research (CIO), Energy and Sustainability Research Institute Groningen (ESRIG), University of Groningen, Groningen, Netherlands*

[2]*Joint International Research Laboratory of Atmospheric and Earth System Sciences, School of Atmospheric Sciences, Nanjing University, Nanjing, China*

[3]*Meteorology and Air Quality, Wageningen University and Research Center, Wageningen, Netherlands.*

[4]*Institute for Atmospheric and Earth System Research/Physics, Faculty of Science, University of Helsinki, Finland*

[5]*ICOS Carbon Portal, Lund University, Geocentrum II, Sölvegatan 12, 22362 Lund, Sweden*

[6]*Institute for Marine and Atmospheric Research, Utrecht University, Utrecht, the Netherlands*

Corresponding author: Huilin Chen (huilin.chen@rug.nl)



**Abstract**: Carbonyl sulfide (COS) is a promising tracer for the estimation of terrestrial ecosystem gross primary production (GPP). However, understanding its non-GPP related sources and sinks, e.g., anthropogenic sources and soil sources and sinks, is also critical to the success of the approach. Here we infer the regional sources and sinks of COS using continuous *in-situ* mole fraction profile measurements of COS along the 60-m tall Lutjewad tower (1 m a.s.l., 53°24'N, 6°21'E) in the Netherlands. To identify potential sources that caused the observed enhancements of COS mole fractions at Lutjewad, both discrete flask samples and in-situ measurements in the province of Groningen were made on a mobile van using a quantum cascade laser spectrometer (QCLS). We also simulated the COS mole fractions at Lutjewad using the Stochastic Time-Inverted Lagrangian Transport (STILT) model combined with emission inventories and plant uptake fluxes. We determined the nighttime COS fluxes to be -3.0 ± 2.6 pmol m$^{-2}$ s$^{-1}$ using the radon-tracer correlation approach and Lutjewad observations. Furthermore, we identified and quantified several COS sources, including biodigesters, sugar production facilities, and silicon carbide production facilities in the province of Groningen. Moreover, the simulation results show that the observed COS enhancements can be partially explained by known industrial sources of COS and $CS_2$, in particular from the Ruhr valley (51.5°N, 7.2°E) and Antwerp (51.2° N, 4.4° E) areas. The contribution of likely missing anthropogenic sources of COS and $CS_2$ in the inventory may be significant. The impact of the identified sources in the province of Groningen is estimated to be negligible to the observed COS enhancements. However, in specific conditions, these sources may influence the measurements in Lutjewad. These results are valuable for improving our understanding of the sources and sinks of COS, contributing to the use of COS as a tracer for GPP.

Keywords: Carbonyl sulfide, anthropogenic source, vertical profile, nighttime uptake

## 1 Introduction

Interest in the budget of carbonyl sulfide (COS) has grown over the last decade due to the close relation of COS and carbon dioxide ($CO_2$) vegetative uptake. The two gases follow a similar uptake pathway from the leaf boundary layer up to the site of reaction in the plant (Stimler et al., 2010). COS therefore provides a means to separate the concurrent uptake of gross primary productivity (GPP) and respiration flux of $CO_2$ (Campbell et al., 2008; Montzka et al., 2007). Those individual fluxes can otherwise not be measured directly at scales larger than the leaf scale. Besides the interest in COS as a tracer for GPP, COS is also of interest in the stratosphere as it plays a role in the formation of the stratospheric sulfate aerosol layer, which has an overall cooling effect to the Earth's climate (Brühl et al., 2012).

On average, mole fractions of COS in the atmosphere range between 350 and 550 parts per trillion (ppt) globally. The vegetative uptake of COS is the largest sink in the atmospheric COS budget, followed by uptake by soils (Berry et al., 2013; Whelan et al., 2018). The main sources of COS are anthropogenic emission, the ocean, wetlands and biomass burning. Anthropogenic emissions of COS can be either direct emissions of COS (e.g. coal combustion, aluminum smelting, pigment and paper industry), or indirect through emissions of $CS_2$ (e.g. rayon production, agricultural chemicals and tire wear), which can be oxidized to COS (Zumkehr et



al., 2018). Unfortunately, the current COS budget has large uncertainties, and lacks COS sources to balance the sinks, mainly due to uncertainties in the contribution of the tropical ocean and anthropogenic emissions (Whelan et al., 2018).

The long-term COS record presented by Montzka et al. (2007) gave insight into the seasonality of COS mole fractions: it showed that in the northern hemisphere the COS mole fraction is largely influenced by uptake by the biosphere, and by oceanic emissions in the southern hemisphere. Those measurements were made using discrete flask samples (1 to 5 samples per month) that were analyzed by a gas chromatographic mass spectrometer. Optical instruments
that are capable of making high-frequency (1 to 10 Hz) *in-situ* simultaneous measurements of COS and $CO_2$ (Stimler et al., 2009) are available, e.g. a quantum cascade laser spectrometer (QCLS). This creates opportunities to advance our understanding of the COS sources and sinks, through flux measurements using the eddy-covariance technique and soil and branch chamber measurements (Berkelhammer et al., 2014; Commane et al., 2015; Kitz et al., 2017; Maseyk et
al., 2014; Sun et al., 2018; Vesala et al., 2022; Wehr et al., 2017; Yang et al., 2018), and through atmospheric mole fraction measurements within the continental and marine boundary layer (Belviso et al., 2016, 2020; Commane et al., 2013; Kooijmans et al., 2016; Lennartz et al., 2017). Moreover, these measurements enabled the collection of in-situ data on a mobile van, which made it possible to identify COS sources directly at their emission sites.

This study aims to investigate the processes that impact the atmospheric COS mole fractions at Lutjewad and to infer the influence of local COS sources on the Lutjewad measurements. This has been realized with continuous atmospheric mole fraction observations of COS, $CO_2$ and carbon monoxide (CO) at the 60 m tall tower, and with discrete flask and continuous *in-situ*
measurements of COS, $CO_2$, $CH_4$, $N_2O$ and CO on a mobile van in the province of Groningen in the Netherlands. Moreover, atmospheric COS and $CO_2$ mole fractions at Lutjewad were simulated for the period of January and February 2018, using the Stochastic Time-Inverted Lagrangian Transport (STILT) model. Finally, we estimated nighttime COS ecosystem fluxes and anthropogenic COS emissions from identified local sources based on atmospheric mole
fraction measurements of COS.

## 2 Methodology

### 2.1 Measurement sites

#### 2.1.1 Stationary measurements

Profile measurements were performed at the Lutjewad atmospheric monitoring station in the Netherlands (53°24'N, 6°21'E). The Lutjewad station is located at the north coast of the
Netherlands in front of the Wadden Sea (largely consisting of tidal mud flats). The first kilometer towards the north is covered by salt marshes. Towards the south, the area is used for agriculture. Much of the land in the area is reclaimed from the sea with the use of dikes since the 15[th] century. The agricultural land around the Lutjewad station has been reclaimed from the Waddensea in the 19[th] and early in the 20[th] century; therefore, the soil consists of clay that





originates from the sea. The station is located next to the dike (which is 7 m high) of the Waddensea and consists of a 60 m tall tower. The area is sparsely populated: the closest village is Hornhuizen (~200 inhabitants) at a distance of 1.3 km towards the south; the closest city is the city of Groningen (~200.000 inhabitants) at a distance of 25 km towards the southeast. 10

km towards the west of the station is a small ferry port. Farmlands around the measurement station are planted with seed potatoes, sugar beets and winter wheat. 40 km towards the southeast is an aluminum smelting factory (Damco Aluminium; 53°18' N, 6°58' E) which lies within the Delfzijl/Farmsum industrial area. Regionally, there are several aluminum and chemical facilities at 250 km distance in the German Ruhr-area (e.g. Trimet Aluminium, Hydro

Aluminium), that may be a source of COS.

*2.1.2 Mobile flask and in-situ measurements*

Several facilities were investigated for their potential COS emissions in the surroundings of

Lutjewad, including both known COS emitters from literature, such as coal-related industries (Campbell et al., 2015; Zumkehr et al., 2018), and potential new sources, such as organic waste treatment plants (Aston & Douglas, 1981; Smet et al., 1998). These locations and their source types are summarized in Table 1, with their locations shown in Figure 1.

*Table 1: possible sources of COS according to the retrieved literature.*

| Location | Source type | Coordinates |
|---|---|---|
| Eemshaven - RWE coal fired powerplant | Fossil fuels | 53.44°N, 6.86°E |
| Grijpskerk - GasUnie facilities | Fossil fuels | 53.27°N, 6.31°E |
| Delfzijl - ALDEL DAMCO aluminium facilities | Aluminium smelting | 53.31°N, 6.98°E |
| Farmsum - Teijin Aramid B.V. facilities | Rayon production | 53.32°N, 6.96°E |
| Groningen - ATTERO facilities | Waste | 53.20°N, 6.62°E |
| Hoogkerk – Cosun Beet (SuikerUnie) facilities | Sugar production, waste | 53.21°N, 6.50°E |
| Groningen - agricultural fields | Ploughing | Various |

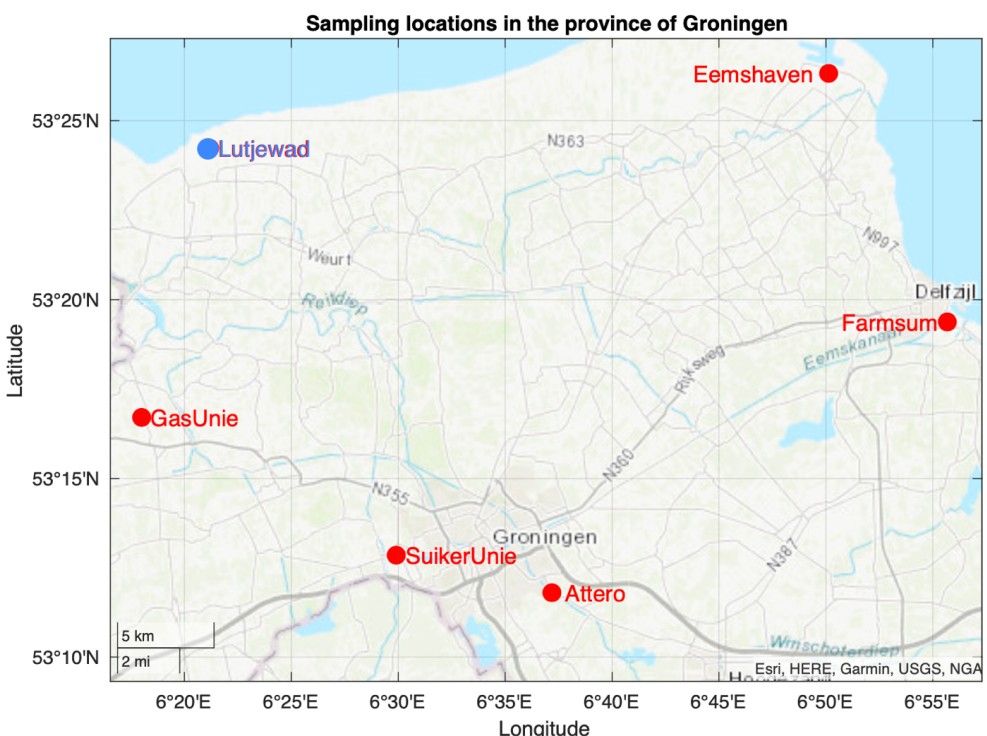

*Figure 1: location of Lutjewad and of the sampling locations in the province of Groningen (NL). Only the locations where emissions were detected will be described in the text.*

### 2.2 Measurements of COS, CO₂ and CO

*2.2.1 Stationary measurements*

A QCLS was used to measure dry mole fractions of COS, $CO_2$, CO and $H_2O$ at different heights of the Lutjewad tower between 2014 and 2018 (Table 2). The measured data were first presented in Kooijmans et al. (2016; Fig. 12) for the period between August 2014 and April 2015. The setup of the QCLS is described in detail in Kooijmans et al. (2016). In summary, the QCLS was sampling air from different heights (see Table 2) and the different sampling lines (Synflex Decabon or Teflon) were switched with a multi-position Valco valve (VICI; Valco Instruments Co. Inc.). The sampling time differed per period (Table 2). A reference cylinder was measured every half hour to correct for instrument drift and to calibrate the measurements to the common scales. Specifically, the reference cylinders were calibrated against two NOAA/ESRL standards for COS (NOAA-2004 scale) and $CO_2$ (WMO-X2007 $CO_2$ scale) at the University of Groningen (Kooijmans et al., 2016). The measurements had to be corrected for a leaking solenoid valve for the period between August 2014 and January 2015. This was done by comparing the $CO_2$ measurements with measurements from a collocated cavity ring-down spectrometer (Picarro Inc. model G2401-m) and applying a similar dilution factor to all gas species (see details in Kooijmans et al., 2016). A target cylinder was measured once every



hour in all periods except for the measurements in Lutjewad in January – February 2018. Kooijmans et al. (2016) gave an overview of all uncertainty contributions that are relevant for obtaining accurate and precise COS mole fractions; that is, the repeatability of the NOAA scale (2.1 ppt), calibration of reference standards and ambient air samples (2.8 ppt), water vapor

correction (2.9 ppt) and measurement precision. The measurement precision (defined as the standard deviation over minute-averaged target cylinder measurements after drift correction with reference measurements) has changed over the years; an overview of the average precision is given for the 2014-2015 period in Table 2.

Field standards are calibrated against NOAA standards in the laboratory before and after each measurement period to test for drift in the cylinders. The COS mole fraction measurements of nine cylinders are available, and five cylinders changed less than 2.5 ppt/year, two cylinders decreased by ~10 ppt/year and 2 cylinders decreased by ~30 ppt/year. The four cylinders that drifted more than 10 ppt/year were not used as reference cylinders in the data processing. All

of the cylinders were uncoated aluminum cylinders, which, according to experience at NOAA, are more prone to COS mole fractions drift than Aculife treated aluminum cylinders.

In Lutjewad, besides the *in-situ* measurements, we also measured flasks that were sampled at 60 m between December 2013 and February 2016 with an average of four samples per month.

81% of the flask samples were taken at noon. For a detailed description of the measurement procedure see Kooijmans et al. (2016). The flask measurements of COS mole fractions were used together with the *in-situ* measurements in Lutjewad to construct a seasonal fit to the data. We constructed a seasonal fit to the 60 m COS and $CO_2$ mole fractions from Lutjewad. The non-linear least squares fit of COS mole fractions is shown in Figure S1 of the supplementary

material and details are explained in the accompanying text.

*Table 2: Measurement periods at the Lutjewad site with an overview of the measurement heights, sampling time and one-minute measurement precision based on target cylinder measurements (in Jan. – Feb. 2018 in Lutjewad no target measurements were made).*

| Location and Period | Measurement heights [m] | Sampling time per height and frequency | Precision [ppt] |
|---|---|---|---|
| *Lutjewad, The Netherlands* | | | |
| Aug. 2014 – Apr. 2015 | 7, 40, 60 | Two times 8 min., every hour | 5.3 |
| Jan. – Feb. 2018 | 60 | Two times 27 min, every hour | - |

*2.2.2 Mobile flask and in-situ measurements*

The mobile and in-situ investigation of the sources described in Section 2.1 was performed in September and October 2019. Firstly, discrete samples were collected in flasks and analyzed

on a QCLS, which allowed the simultaneous analysis of COS, CO, $CO_2$, $CH_4$ and $N_2O$. Afterwards, a van was equipped as a mobile sampling station to realize *in-situ* continuous analysis, to allow immediate detection of COS enhancements. A QCLS was placed in the inside of the van, where electricity was supplied by three 115Ah 12V lead acid batteries via a



MeanWell TS700 inverter. The instrument pulled air through a sampling line, with its inlet placed on the top of the vehicle. The sampling line was equipped with a reverse cup as rain guard and a Nafion dryer to remove most water vapor from the air samples. During sampling, GPS live data synchronized with the QCLS time log were collected. Generally, this method

allowed a real time investigation of the interested areas, enabling the understanding of spatial distribution of trace gases concentration.

### 2.3 Nighttime ecosystem flux in Lutjewad

Nighttime fluxes of COS and $CO_2$ are estimated for the Lutjewad area based on the radon-tracer method, similar to the calculation of nighttime fluxes in Hyytiälä by Kooijmans et al. (2017). Measurements of $^{222}Rn$ can be used to calculate fluxes of other gases, because $^{222}Rn$ is produced in the soil with a constant rate and it diffuses through the soil into the air. Once it is in the atmosphere, it is only affected by radioactive decay and by the effect of atmospheric mixing.

The nighttime mole fractions of gases get either enriched (in the case of dominant sources) or depleted (in the case of dominant sinks) in a shallower nocturnal boundary layer compared to the daytime boundary layer. This means that, when the $^{222}Rn$ exhalation rate ($F_{Rn}$) is known, the surface fluxes of another gas (in this case of COS ($F_{COS}$) and $CO_2$ ($F_{CO2}$)) can be determined from the mole fraction changes of the gas ($\Delta COS$ and $\Delta CO_2$) over the night, relative to that of

$^{222}Rn$ ($\Delta^{222}Rn$): e.g. $F_{COS} = F_{Rn} * (\Delta COS/\Delta^{222}Rn)$ (Belviso et al., 2013, 2020; Schmidt et al., 1996; van der Laan et al., 2009). $F_{Rn}$ was determined for the Lutjewad area in different measurement and modelling studies of which an overview is given in van der Laan et al. (2016). In these studies, $F_{Rn}$ varied between 2.3 and 5.1 mBq m$^{-2}$ s$^{-1}$. We will use the average over these studies, 3.7 mBq m$^{-2}$ s$^{-1}$, with a standard deviation of 1.2 mBq m$^{-2}$ s$^{-1}$. The $^{222}Rn$ measurements

in Lutjewad are made with an ANSTO dual-flow loop two-filter detector (Whittlestone & Zahorowski, 1998). Details about the measurement procedure are described in van der Laan et al. (2009). COS and $CO_2$ fluxes are only calculated for nights when at least 7 data points are available, where the $R^2$ values between $^{222}Rn$ and COS ($CO_2$) mole fractions are larger than 0.4 (0.5) and where the standard error of the flux (based on the uncertainty of the slope between

$^{222}Rn$ and COS or $CO_2$ mole fractions) is smaller than 4 pmol m$^{-2}$ s$^{-1}$ (COS) and 1.5 μmol m$^{-2}$ s$^{-1}$ ($CO_2$). Furthermore, the uncertainties of the radon-tracer method largely result from the uncertainty of $F_{Rn}$. The flux uncertainty is therefore calculated as the quadrature sum of the uncertainty on the slope and of $F_{Rn}$ (1.2 mBq m$^{-2}$ s$^{-1}$).

### 2.4 Simulations of COS mole fractions


To understand the influence of natural and anthropogenic COS sources on the concentration measurements at Lutjewad, atmospheric transport simulations were performed to obtain COS mole fractions at the station. The simulation covers the period from January to February 2018,

given the availability for both observations and models data for such dates. To simulate the enhancements from the COS background concentrations, the Stochastic Time-Inverted Lagrangian Transport (STILT) model (Lin, 2003), driven by ECMWF-IFS operational analysis, was combined with COS biosphere and soil fluxes from the Simple Biosphere model, version 4 (SiB4) (Kooijmans et al., 2021) and with the anthropogenic emission database by Zumkehr



et al. (2018). The COS background was estimated using the end point of the STILT model trajectories in the analysis domain and the derived 3D concentration fields from the Transport Model 5 – Four-Dimensional Variational model (TM5-4DVAR) inversions (Ma et al., 2021). The STILT model uses meteorological data to determine the origin of air parcels influencing the measurements of a defined location at a specific point in time. Each simulation run releases 100 particles from the Lutjewad station, at a height of 60 m. The transport of these particles is reconstructed within the selected domain (latitude 34.0°N-73.5°N, longitude 20.0°W-45.5°E, to cover Europe), in three-hours timesteps over a desired period back in time. Depending on the number, the location and the height of the particles, the model computes footprints in ppm / ($\mu$mol m$^{-2}$ s$^{-1}$), at a 0.1°x0.1° resolution, indicating the influence of specific areas on the final measurements. An example is shown in Figure 2a. The resulting footprint value gets smaller for each timestep back in time, and is thus less influential for the simulated concentration at the receptor. In this analysis, footprints are typically negligible after 8 to 9 days. Therefore, the simulation timespan is set on 10 days to confidently cover all the potentially significant footprint values.

The SiB4 and the anthropogenic emission databases include gridded COS fluxes (pmol m$^{-2}$ s$^{-1}$) and were interpolated to grids of 0.1°x0.1° to match the STILT footprints. Biospheric COS fluxes are defined for each 3-hours timestep, depending on time of the day and seasonality. In the considered period, these fluxes are negative, mainly due to COS uptake by soils. Anthropogenic fluxes are assumed to be constant over time and include both direct and indirect COS emissions. The anthropogenic COS emissions map is shown in Figure 2b as an example. The indirect emissions are accounted as $CS_2$ fluxes. The conversion of $CS_2$ to COS is computed with two different scenarios, considering a 3-days-exponential (Khan et al., 2017) and a 10-days-exponential conversion rate (Ma et al., 2021), with a reaction yield of 0.87 (Ma et al., 2021; Zumkehr et al., 2018). Therefore, the indirect COS fluxes are calculated for each timestep $i$ back in time (maximum 240 hours or 10 days) as described by Equation 1. All COS fluxes are then multiplied by footprint values $f$ to obtain the relative COS contributions, $\Delta COS_i = F_i * f$. Consequently, the COS enhancement at the receptor $\Delta COS$ consists of the contribution from the biospheric fluxes $\Delta COS_{bio}$, the contribution from the constant direct anthropogenic emissions $\Delta COS_{ant}^{dir}$ and the contribution from time varying indirect anthropogenic fluxes $\Delta COS_{ant}^{ind}$.

$$F_{ant,i}^{ind} = F_{CS_2} * \left(1 - e^{-\frac{i}{\tau}}\right) * 0.87, \text{with } \tau = 3 \text{ days}, 10 \text{ days (Equation 1)}$$

$$\Delta COS = \Delta COS_{bio} + \Delta COS_{ant}^{dir} + \Delta COS_{ant}^{ind} \quad \text{(Equation 2)}$$

As stated earlier in the text, the background was estimated using the endpoint of the particles in the STILT model and the 3D concentration fields from the TM5-4DVAR simulations. These are geospatially defined by a 6° x 4° x 1km (longitude x latitude x altitude) grid, where each box of this grid is related to a specific COS concentration. The endpoint of each particle's trajectory within this grid was therefore associated to its respective concentration (see Figure S2). For each timestep $t$, the COS background is calculated as the average of the COS concentrations over the 100 particles endpoints of the STILT model. The product between





gridded footprints and fluxes, instead, yields the contribution of each location on the COS enhancements over the background in Lutjewad, in ppt (see Figure 2c).

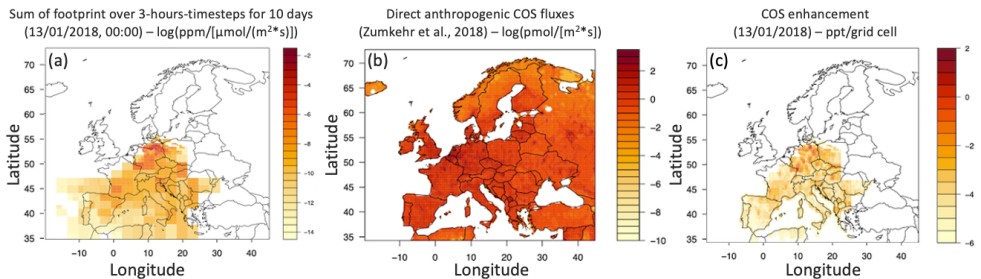

*Figure 2: reported in logarithmic scales: (a) example of localized footprint values resulting*
*from the STILT model simulations, summed over 10 days before the starting timestep (13/01/2018, 00:00), (b) the localized direct COS sources according to Zumkehr et al. (2018), (c) the modelled enhancement resulting from the product of (a) and (b), identifying the sources influencing Lutjewad in North-Eastern Germany (the minimum for this scale was set at -2 for clarity purposes).*

Ultimately, the total COS molar fraction simulation of each timestep can be calculated using Equation 3, over 3-hours timesteps, for the months of January and February 2018:

$$C_{COS} = B_{COS} + \Delta COS \quad (Equation\ 3)$$

Where $C_{COS}$ is the total COS molar fraction, $B_{COS}$ the COS background, $\Delta COS$ is the COS enhancement (or depletion, for $\Delta COS_{bio}$) associated with fluxes calculated with Equation 2.
In addition, $CO_2$ molar fractions were simulated using the STILT single-site scoped viewer (Karstens et al., 2022). This tool combines STILT simulations with anthropogenic $CO_2$ emissions categorized by sector from the EDGARv4.3 inventory (Janssens-Maenhout et al.,
2017) and biospheric $CO_2$ fluxes from the Vegetation Photosynthesis and Respiration Model (VPRM) (Mahadevan et al., 2008).

## 3 Results

### 3.1 Sources and sinks by wind directions during stationary measurements

Figure 3 shows the deviation of the COS, $CO_2$ and CO mole fractions from their seasonal cycle for the Lutjewad site against wind direction. A negative (positive) deviation (e.g., $COS_{7m}$ − $COS_{seas.} < 0$) is indicative of a sink (source) that is not represented by the seasonal cycle.
Typically, there is a difference in signals between daytime and strongly stable nights, especially for deviations of 7 m mole fractions (left plots in Figure 3). No large COS deviations are observed for daytime data and weakly turbulent nights (when the temperature gradient between 60 and 7 m is lower than 0.75 °C), apart from a decrease of ~ 15 ppt with wind from the east (see Figure 3a-b). For nighttime data with strongly stable conditions, we observe larger
deviations from the seasonal cycle. For 7 m deviations we generally observe the largest depletions in COS from eastern wind and southwestern wind, which is with wind from inland





(wind directions between 50 – 300º). However, no clear depletions are observed with wind directions from the south. For 60 m (right plots in Figure 3) we also find COS to be depleted in eastern wind directions (Figure 3b), and see weak depletions for winds from the southwest. COS mole fractions were substantially lower at all heights in a period of a few days between 1 and 8 September 2014 (not shown).

For $CO_2$ and CO, we observe elevations from the seasonal cycle for both day and night. The elevations span the range of wind directions for which air originates from land. The $CO_2$ mole fractions are further enhanced in the nighttime. The CO elevations are similar for day and nighttime, apart from a peak at 200 degrees, which is higher during strongly stable conditions.

Peaks at night do not necessarily point to larger sources or sinks in a certain direction, but could originate from a few nights with strongly stable conditions that drive large changes in mole fractions and that have a relatively large influence on the averages. The binned averages of the strongly stable nighttime conditions are more prone to such peaks because these data represent less data (332 data points) than the weakly stable nights (1269 data points).





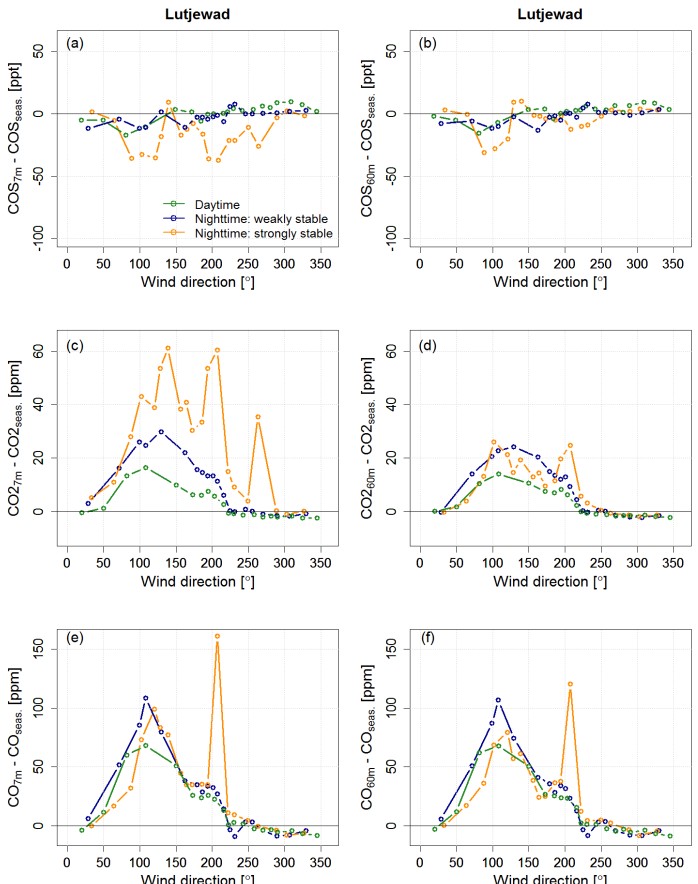

*Figure 3: Deviation of 7 m (left) and 60 m (right) mole fractions of COS (a,b), CO₂ (c,d) and CO (e,f) from their seasonal cycle in Lutjewad. Data are separated between daytime (solar elevation angle > 0°; green) and nighttime (solar elevation angle < 0°), where nighttime data are divided over weakly (blue) and strongly (orange) stable nights, which are separated based on the temperature difference between 60 and 7 m being smaller or larger than 0.75 °C.*

### 3.2 Estimate of nighttime COS and CO2 fluxes

Figure 4 shows the nighttime fluxes of COS and $CO_2$ in Lutjewad based on the radon-tracer method. Most of the derived COS fluxes are negative, implying COS sinks at the surface. Occasionally, there are positive fluxes, which coincide with periods in which we observe COS spikes after ploughing (see Kooijmans, 2018). The median nighttime COS flux is $-3.0 \pm 2.6$ pmol m$^{-2}$ s$^{-1}$ (excluding the positive fluxes), with $-2.9 \pm 2.2$ pmol m$^{-2}$ s$^{-1}$ from August to November 2015 and $-7.2 \pm 2.8$ in April 2015. The nights with COS emissions have an average COS flux of $+3.5 \pm 2.1$ pmol m$^{-2}$ s$^{-1}$. Nighttime $CO_2$ fluxes decrease from August to December, then increase in January and reach highest $CO_2$ fluxes in April (note that $CO_2$ fluxes from May-July are not available).



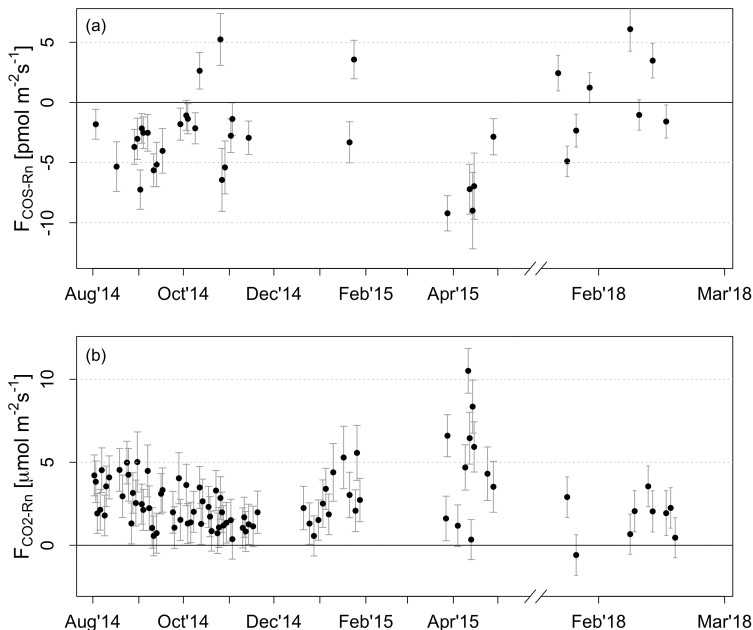

*Figure 4: Nighttime fluxes of COS (a) and CO₂ (b) in Lutjewad based on the radon-tracer method. Note that the x-axis jumps from April 2015 to February 2018.*

### 3.3 Modelled and observed COS and CO₂ mole fraction

The period of January and February 2018 is characterized by a few episodes of increased COS mole fractions that sometimes last for a few hours, but also extend to a few days (Figure 5). This period was characterized by cold weather (air temperature < 0 °C), which allowed ploughing activities with heavy machinery in the agricultural fields surrounding the station. At 60 m, we observed COS elevations in the order of hundreds of ppt above the background mole fraction over a period of a few days. $CO_2$ and CO molar fractions are also elevated when COS is higher. $CO_2$ and CO mole fractions are strongly correlated in this period ($R^2 = 0.94$) and the ratio of CO to $CO_2$ elevations in this period is 5.3 ppb ppm$^{-1}$. COS mole fractions are not as strongly correlated with $CO_2$ and CO ($R^2 = 0.50$ and 0.48, respectively).

The observations of COS and $CO_2$ in the period of January and February 2018 were further investigated, using the simulations described in Section 2.4. Figure 5 and Figure 6 show the comparison between the modelled results and the measurements in Lutjewad for January and February 2018. The model generally reproduces the measurements trend well for both species. For $CO_2$, the average difference between measurements and modelled values was 3.6 ± 5.4 ppm. The model captures the $CO_2$ enhancements in January 22-28 (Period 1, $R^2 = 0.74$), January 30-31 (Period 2, $R^2 = 0.88$), February 5-11 (Period 3, $R^2 = 0.61$) and February 12-15 (Period 4, $R^2 = 0.82$), although generally it slightly underestimates the total molar fraction (see Figure 5, Figure 6 as well as Figure S3 and Figure S4 in the supplementary information section). Figure





7 shows the contribution of background, biosphere and anthropogenic emissions to the final results for both gases. Anthropogenic emissions represented the biggest contributors to the deviations from the background for both gas species. As expected, the biospheric influence results in emissions for $CO_2$, due to the respiration process which dominates plant behavior in

winter. In contrast, the biospheric contribution to COS molar fraction results in depletion of this gas species, which can be attributed to soil uptake. Four periods when either COS or $CO_2$ showed significant enhancements from the background were selected within the investigated timeframe. According to the model, most of the enhancements can be attributed to industry in the Ruhr area (Period 4) and the Antwerp-Rotterdam region (Periods 1, 2 and last part of Period

3). Interestingly, the Ruhr area is also responsible for the overestimation occurring on February 19 for both species. For Period 1, 2 and 4 the model estimates roughly between 51% and 68% of the measured enhancements. On the other hand, Period 3 is related to the lowest $R^2$ value and to the highest underestimation, simulating just around 32% of the measured enhancements. This is the only period related to eastern footprints in the selected timeframe. With regard to

COS, it is clear that the model generally shows a slight overestimation of its molar fraction, with an average difference between measurements and modelled values of -4.5 ± 26.9 ppt. The model is generally less accurate in reproducing COS mole fractions when compared to its $CO_2$ performance (for a regression analysis of simulation against observations, see Figure S3 and Figure S4 in the supplementary information section). However, the model still captures 61% of

the enhancements of Period 4 ($R^2 = 0.70$), which the STILT model attributes to the Ruhr area. Moreover, the model captures singular peaks related to Ruhr area's emissions in Period 1 (over the whole period, $R^2 = 0.23$). Furthermore, it reproduces the trends of the enhancements in the second part of Period 3 (February 8-11). This period is related to a mixed southern and eastern footprint, which ascribes this share of enhancements to the Antwerp-Rotterdam area and to

paper production locations in northern Germany. Nonetheless, severe underestimations occur persistently between February 7-10 and as singular events around February 6 and 17. The largest underestimation of COS reaches around 254 ppt on February 7. As stated earlier in the text, Period 3 and February 17, unlike most of the other periods, are characterized by eastern footprint outputs, followed by high footprint values close to the Lutjewad area. Altogether, this

suggests that the emissions of both COS and $CO_2$ east of Lutjewad may be underestimated. Noticeably, the highest $CO_2$ underestimations, occurring between February 9 and 10 and reaching up to 26.9 ppm, are related to mostly southern influences, but still showing high influences from the Lutjewad surroundings.



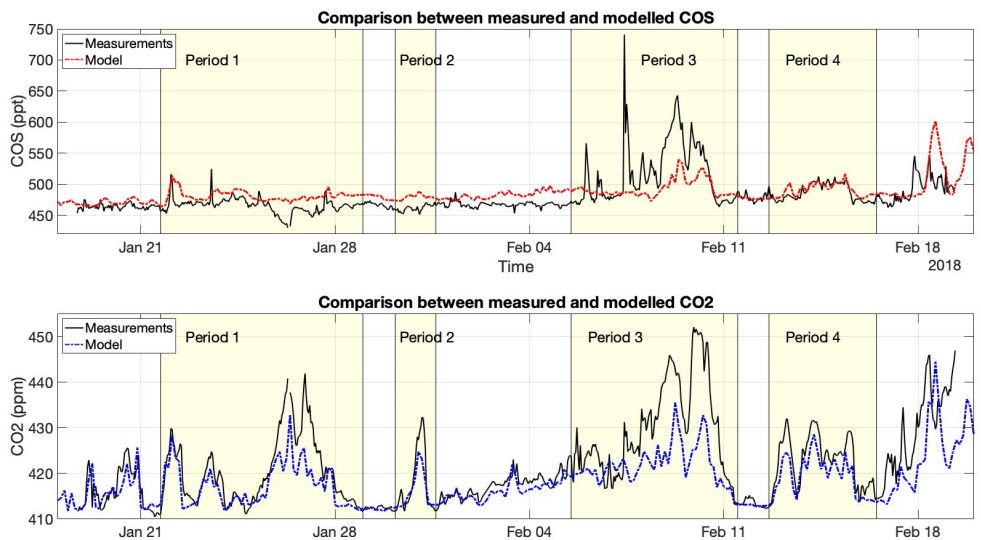

*Figure 5: modelled and observed mole fractions of COS and $CO_2$ in Lutjewad (60 m a.g.l.) in January and February 2018. The periods of interest in this time frame are highlighted in yellow: during these time intervals, $CO_2$ and/or COS enhancements were measured.*

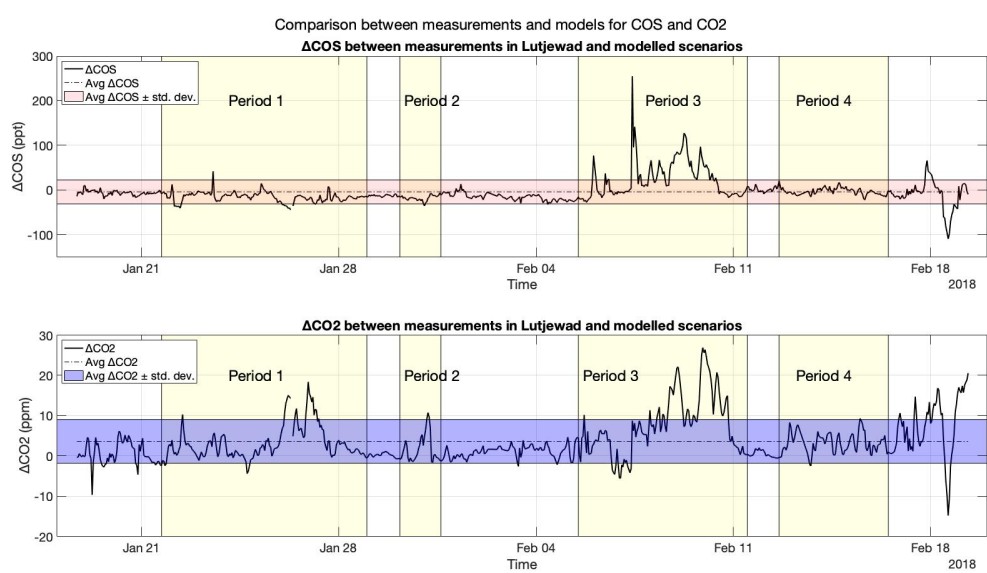

*Figure 6: difference between modelled and observed COS and $CO_2$ mole fractions. The red and blue shaded areas include the values lying between the average difference between measurements and models ± the standard deviation of this difference.*



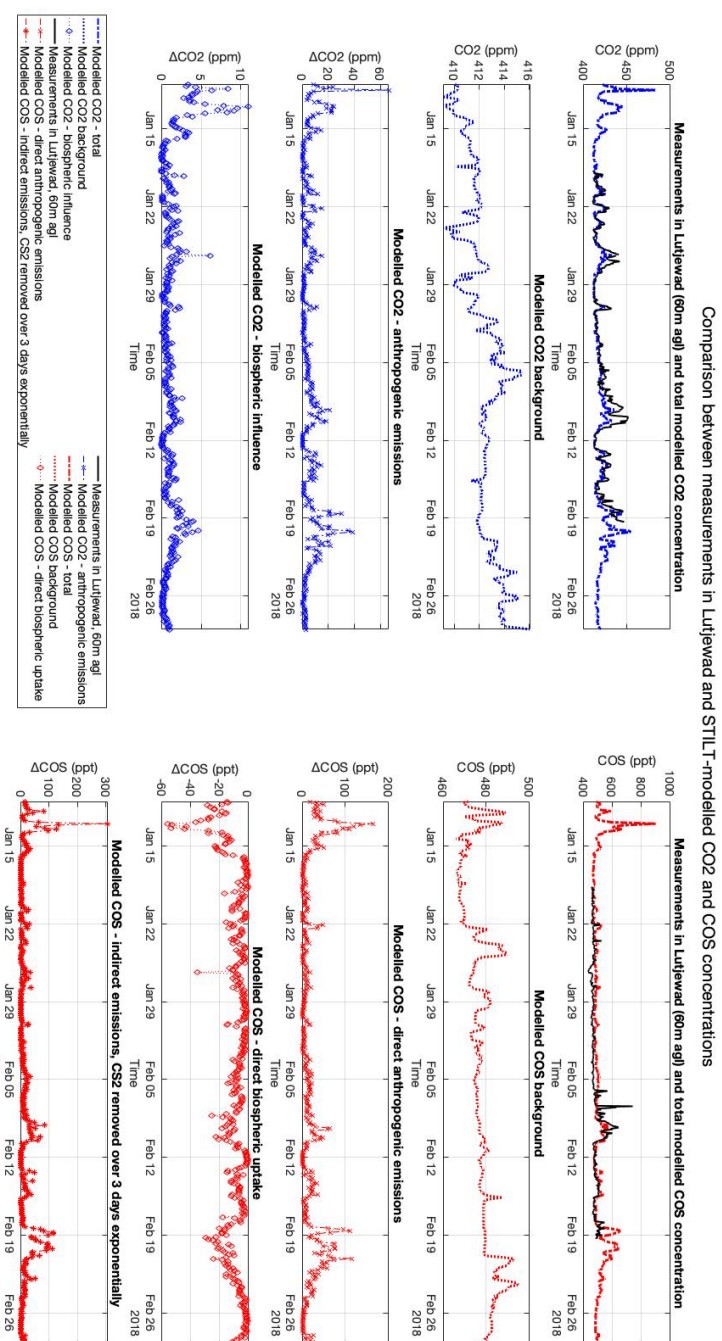

*Figure 7: mole fractions of CO₂ and COS, showing the contributions of background, biosphere and anthropogenic emissions. The top plots show the difference between modelled results and measurements for both gases. The indirect emissions of COS in this figure were computed assuming a 3-days exponential conversion of CS₂ to COS.*



### 3.4 Discrete samples and in-situ measurements

During the sampling activities using flasks and a mobile van, described in Section 2.2.2, COS
sources were identified. In particular, emissions were found at the SuikerUnie facilities
(53.2°N, 6.5°E), at the ATTERO facilities (53.2°N, 6.6°E) and at coal- and aluminum-related
industries in Eemshaven (53.4°N, 6.8°E) and in the Delfzijl/Farmsum area (53.3°N, 6.9°E) (see
Figure 1). Given the southeasterly wind direction during sampling, it was not possible to
separate the contribution of each company in Farmsum to the measured mole fractions.
Therefore, these results will be reported by the name of the industrial facilities: *ChemiePark*.
The only company that could be easily isolated in the area was ESD-SiC (53.3°N, 7.0°E), a
silicon carbide producer (see Figure 8a,b). This company is known to be related to occasional
explosive events (Dagblad van het Noorden, 2018; ESD-SiC, 2018; The Northern Times,
2018), which will be discussed later in the text. Discrete sampling was performed in
Eemshaven, where industries and energy plants based on fossil fuels can be found, and at the
ATTERO facilities for waste treatment and biogas production. The results of these samples are
presented in Table S1 and Table S2 of the supplementary information section. Among these
results, COS enhancements between tens and about 100 ppt were measured at the waste disposal
site and at the biodigesters of ATTERO. SuikerUnie facilities are also producing biogas from
the sugar treatment leftovers and at this site COS mole fractions went up to 1.8 ppb, almost 1.3
ppb above the background values. These findings are of particular interest, as will be described
in Section 4.1. In-situ measurements from a mobile van were performed at the fields nearby the
Lutjewad station at the end of October 2019, while the area was being ploughed. In this
occasion, no COS enhancements were detected from ploughing activities. The results of
continuous measurements showing COS enhancements are reported in Table 3. The fluxes for
the in-situ measurements were calculated with a Gaussian dispersion model after Csanady et al.
(1973), using COS mole fractions, distance from the source and wind speed. The errors were
estimated performing a Monte Carlo simulation, similarly to Bakkaloglu et al. (2021). The COS
enhancements from the background were chosen from a uniform distribution within the
observed enhancements range. Distance from the source and wind speed were selected from a
normal distribution centred at the estimated distance and average wind speed. The estimated
wind speed determined the stability class for the Gaussian dispersion model for each specific
run of the Monte Carlo simulation. For some of these sources, such as SuikerUnie, biodigesters
and industries in Farmsum, co-emissions of COS with CO, $CO_2$, $CH_4$ and $N_2O$ were
occasionally measured. As reported in Table 3, the highest enhancements were measured at the
ChemiePark and the related fluxes were consequently estimated in the range of 9369 ± 8582
kg(COS) $a^{-1}$.

*Table 3: summary of COS fluxes obtained with in-situ measurements combined with Monte
Carlo simulations. COS fluxes are reported as both COS and S emission rates. SuikerUnie is a
seasonal factory that runs for about 5 months; thus, the reported yearly emissions should be
considered just a tool to compare the magnitude of different sources of COS when the
companies are active.*



| Source | COS peaks (lowest - highest) | Distance from source | Wind speed | COS emission rate (mean ± std. dev.) | S emission rate (mean ± std. dev.) |
|---|---|---|---|---|---|
| **SuikerUnie** (53.2°N, 6.5°E) | 0.71 - 1.27 ppb | 300 ± 100 m | 7.9 ± 3.9 m s$^{-1}$ | 0.05 ± 0.03 g(COS) s$^{-1}$ 1440 ± 982 kg(COS) a$^{-1}$ | 0.03 ± 0.02 g(S) s$^{-1}$ 769 ± 524 kg(S) a$^{-1}$ |
| **ChemiePark** (53.3°N, 6.9°E) | 1.32 - 6.97 ppb | 400 ± 200 m | 6.4 ± 3.2 m s$^{-1}$ | 0.30 ± 0.27 g(COS) s-1 9369 ± 8582 kg(COS) a$^{-1}$ | 0.16 ± 0.14 g(S) s$^{-1}$ 4986 ± 4581 kg(S) a$^{-1}$ |
| **ESD – SiC** (53.3°N, 7.0°E) | 0.42 - 0.69 ppb | 600 ± 100 m | 6.4 ± 3.2 m s$^{-1}$ | 0.07 ± 0.03 g(COS) s$^{-1}$ 2307 ± 1016 kg(COS) a$^{-1}$ | 0.04 ± 0.02 g(S) s$^{-1}$ 1231 ± 542 kg(S) a$^{-1}$ |

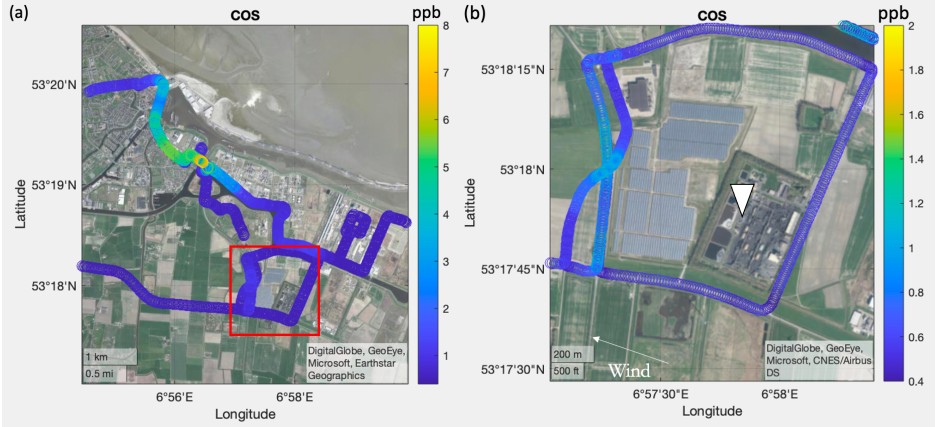

*Figure 8: (a) results of the COS in-situ observations in the Farmsum site. The ESD-SiC area is highlighted with a red square in (a) and shown alone in (b). The emissions at ESD-SiC were strongly correlated with CO and CH$_4$.*

*3.4.1 Influence of observed local sources*

As described in Section 3.3, mismatches were found between measurements at the Lutjewad station and the respective modelled results. Often, these mismatches were related to high influences of areas East of Lutjewad and in the station's surroundings, in particular in the areas of Groningen and North-East Germany. Therefore, we added the fluxes in Table 3 to the available anthropogenic database described in Section 2.4 to check whether they could explain the gap described in Section 3.3. After the implementation of the local fluxes at their respective coordinates, all belonging to the Groningen area, these sources accounted for a total of 93.7 pmol m$^{-2}$ s$^{-1}$. Consequently, COS mole fractions were recalculated using the model described in Section 2.4. The resulting time series was then compared with the results described in Section 3.3, as shown in Figure 9. The additional sources, according to the estimated fluxes, would only have a marginal effect on the final results. Nonetheless, the highest increases occur during the same time periods when the model underestimates the most the results (February 5-10 and, in smaller measure, 17-19), signalling a higher local influence at such occasions.



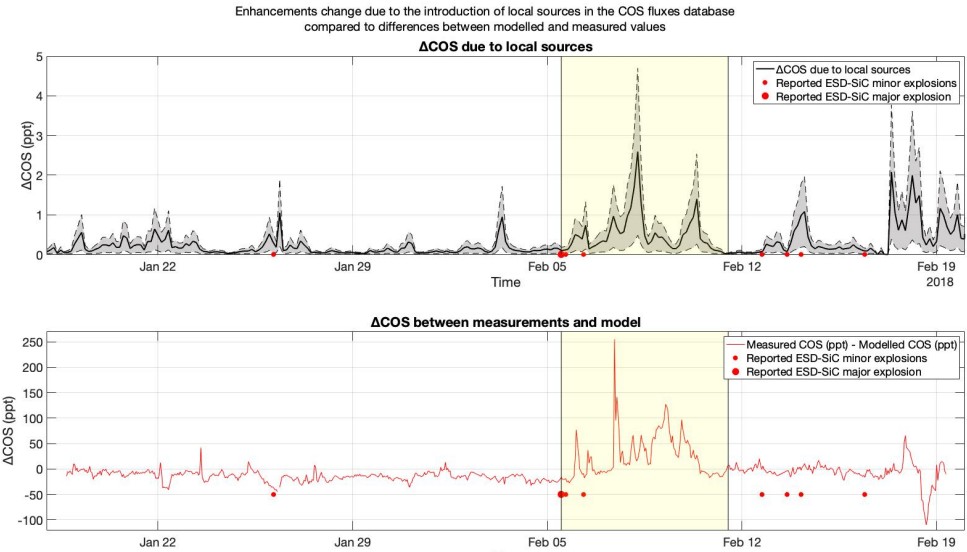

*Figure 9: COS increase in the results after the introduction of local sources in the model, reported as mean contribution (black line) ± the relative standard deviation (grey area). The period when the highest underestimations occur, which coincides to the highest contributions of local sources to the final results, is highlighted in yellow. The red dots show explosive events occurring at ESD-SiC, which will be discussed later.*

To understand how large local emissions would have to be to explain this gap, the highest enhancement from the background (261.7 ppt, on February 7, 9AM) was divided by the sum of local footprints between Lutjewad and the identified sources (52.9-53.4°N, 6.3-7°E) associated to such date. This resulted in an estimated local flux of 148.3 pmol m$^{-2}$ s$^{-1}$. Hypothesizing an even distribution of the sources presented in Table 3 over the same area, they would result in a flux of 2.4 pmol m$^{-2}$ s$^{-1}$. Therefore, the estimated local flux needed to justify the highest measured enhancement in Lutjewad is roughly 2 orders of magnitude higher than the one resulting from the measurements in Table 3. This suggests this peak could only be related to a peculiar event, as will be discussed later in Section 4.1.1. However, the model resolution might have not been high enough to reproduce the dispersion of emissions in such a limited zone. Moreover, it is possible that other sources could be present nearby Lutjewad, or in general in the areas influencing the observations at the tower. Furthermore, the vertical mixing parameter of the model may have been too fast to correctly simulate the plume transport in such a limited area with stable night conditions. Also, possible indirect emissions of CS$_2$ were not considered in this simulation. In other words, a model with a higher resolution and/or a more detailed database would probably produce a different and more accurate estimate for the missing source in the area. Therefore, the number stated above should be considered as a rough estimate.

## 4 Discussion

### 4.1 Anthropogenic sources of COS



The COS enhancements measured in Lutjewad between 2014 and 2018 were firstly attributed to either ploughing activities or other anthropogenic emitters. As reported in Section 3.4, no emissions were found from ploughing activities during this study. However, the measurements

for ploughing activities were rather limited and it is still possible that a natural rapeseed fertilizer (Belviso et al., 2022) or the soil act as a COS source occasionally, depending on soil moisture, temperature, composition and use (Kaisermann et al., 2018; Katayama et al., 1992; Kitz et al., 2017; Maseyk et al., 2014; Whelan et al., 2018). Overall, it remains unclear if ploughing contributed to the measured COS enhancements.

The results presented in Section 3.4 prove the presence of local sources of COS in the province of Groningen. Unfortunately, it was not possible to link the emissions to specific production rates or resources consumption of the observed companies due to lack of information about these parameters. Nonetheless, it is notable that COS emissions were measured from biodigesters, present at the ATTERO and at the SuikerUnie facilities. Biodigesters are currently

not included as sources in the available databases (Campbell et al., 2015; Zumkehr et al., 2018), but the presence of COS has been reported in different food products, such as cheese and cabbage (Aston & Douglas, 1981; Maarse, 1991). Therefore, the role of organic waste as a source of COS could be potentially significant on a global scale and should be further investigated.

According to the footprints obtained using the STILT model, some COS enhancements can be ascribed to known European industrial areas. These include the Ruhr area (Germany), Antwerp (Belgium), Rotterdam-Amsterdam (Netherlands) and, less frequently, North-East Germany, Eastern Europe or the United Kingdom. The Ruhr area, in particular, seems to be almost fully accountable for the enhancements measured between February 14-15 (Figure 5, Figure 6,

Figure 7). However, the mismatch between February 5 and February 10 could not be ascribed to air transport from known sources. In this period, STILT simulations found eastern footprints showing high influences of Lutjewad's surroundings. The influence of local sources would be small according to the available measurements. Nonetheless, as described in the following paragraph, specific exceptional local events could explain the unusual high COS mole fractions

measured during this time interval.

### 4.1.1 Explosions at ESD-SiC

Among the measured local COS sources, ESD-SiC deserves a particular focus. The company produces silicon carbide (SiC) using the Acheson process, which involves high-temperature

furnaces where petroleum coke and silica (sand, $SiO_2$) can react, producing SiC and CO. The reaction between petroleum coke and sand produces low calorific process gas which contains around 1% of sulfur-containing compounds (ESD-SiC, 2022). This gas then undergoes a desulphurization process which removes around 90% of the sulfur compounds (ESD-SiC, 2022). Coke-derived gases have been reported to contain sulfur compounds, including COS

(Ferm, 1957). Zeng et al. (2021) also report significant quantities of $CS_2$ and COS being produced by the thermal-oxidative reaction of sulfur-containing compounds in presence of hydrocarbons. Together with what ESD-SiC explains in their website, this could explain the



observed COS enhancements reported in Table 3. Moreover, this company has been reported to cause nuisance in several occasions with smell or, more noticeably, with explosions (ESD-SiC, 2018; Provincie Groningen, 2018). Local newspapers broadly covered these occurrences: already on January 13, 2015 an explosion covered the villages of Meedhuizen and Tjuchem

(situated about 5 km South-southwest of ESD-SiC) in SiC soot (RTVNoord, 2015). In those dates, no COS enhancements were observed in Lutjewad. However, the footprints calculated for that period suggest that the measured air originated Southwest of the station (ICOS, 2022). Later, frequent explosions occurred in January and February 2018 (Dagblad van het Noorden, 2018; ESD-SiC, 2018; Provincie Groningen, 2018; RTVNoord, 2018; The Northern Times,

2018). Among these, a particularly severe explosion happened on February 5, 2018 at 11.55, which was followed by two smaller explosions, the same day at 15.54 and on February 6[th] at 7.15 (ESD-SiC, 2018). As explained in Section 3.4, the highest COS and $CO_2$ enhancements in Lutjewad, severely underestimated for both species by the modelled results, were found between February 5 and February 10. The footprints related to these dates indicate eastern

origins for the measured air. This finds further confirmation in local newspapers articles which, again, describe easterly winds and soot-related nuisance in villages West of ESD-SiC following the explosions (Dagblad van het Noorden, 2018; RTVNoord, 2018). The measurements for ESD-SiC (Table 3) during a non-explosion occasion and their implementation in the model (Figure 9) would not justify the differences between the modelled results and the measurements.

However, given the results and the information available, the occurrence of these explosion during easterly wind conditions could be the reason behind the enhancements measured in Lutjewad between 2014 and 2018.

### 4.2 Spatial distribution of COS and $CO_2$ sources and sinks

The wind direction analysis in Figure 3 aids in identifying the main sources and sinks of COS in the region of the Lutjewad measurement station, although we have to consider that sources and sinks can balance each other. In general, we find depletions of COS only coming from inland, which is likely driven by terrestrial vegetation and soil. In wind directions from the North, we did not observe a deviation from the seasonal cycle, indicating that the mud flats and

salt marshes are not a strong net source or sink of COS. Still, a source of COS in the salt marshes could be balanced by COS uptake from plants.

The fact that we observe COS depletion at 60 m during daytime is an indication that this is a regional signal rather than a local signal. The depletions of COS that we observe from the

southwest are larger at 7 m under strongly stable nighttime conditions than during daytime and at 60 m, which implies that these depletions are caused by more local sinks of COS. On average, we do not detect a sink from the south, even though this also covers continental air masses, including agricultural land nearby. Vegetative uptake of COS in this wind sector could be balanced by COS sources. The data in Figure 3 mainly represent the autumn and winter months

with only the beginning and end of the growing season. Larger COS depletions can be expected in the summer months if vegetation plays a dominant role in the uptake of atmospheric COS.



The elevated $CO_2$ mole fractions during daytime likely originate from anthropogenic activities, which is substantiated by elevated CO mole fractions in the same range of wind directions. In the nighttime we find $CO_2$ mole fractions to be further elevated than during the daytime, because the effects are amplified in a shallow mixing layer. In both cases, we cannot attribute

these elevations to anthropogenic sources alone because the net ecosystem exchange (NEE) of $CO_2$ can contribute significantly to these elevations. Other tracers, e.g. $^{14}CO_2$, are needed to partition the $CO_2$ elevations into anthropogenic emissions and NEE (Turnbull et al., 2009; van der Laan et al., 2010; Vogel et al., 2010). The wind directions where $CO_2$ enhancements at 7 m shows a peak in the night (200º and 275º) are also the wind directions where COS depletions

are larger, and for one of the two peaks there is also a CO peak (200º). We are not aware of any anthropogenic activity that could lead to depletions of COS and at the same time emit $CO_2$ and CO. The sources and sinks of these gases do therefore not have to be related. We also have to consider that wind directions may differ at 7 and 60 m, especially during the night. Moreover, a few nights with strongly stable conditions and a particular wind direction could have a large

influence on the averages, which would affect all gases and would be detected as a peak.

### *4.3 COS and GPP*

The results presented in Section 3.3 and Section 3.4 underline the relevance of assessing a

thorough regional COS budget in the context of using this gas as a tracer for GPP. From this study, it is clear that both the background molar fraction and the enhancements measured in Lutjewad are influenced by anthropogenic sources. In fact, excursions of COS molar fraction can be ascribed to both local sources and sources located hundreds of kilometers afar from the station, such as the Ruhr area in Germany. A poor assessment of COS sources may lead to

biased findings with regard to COS fluxes estimations, which would therefore mislead the GPP evaluation. With regard to the observed COS enhancements, inverse transport models provide a tool to prevent inaccurate interpretations, or at least to allow a preliminary assessment of possible biases due to the origin of the analyzed air.

### 5 Conclusions

We have inferred the regional sources and sinks of COS using continuous *in-situ* mole fraction profile measurements of COS along the 60-m tall Lutjewad tower (1 m a.s.l., 53°24'N, 6°21'E) in the Netherlands. To identify potential sources that caused the observed enhancements of COS mole fractions at Lutjewad, we have made both discrete flask samples and in-situ measurements

in the province of Groningen on a mobile van using a quantum cascade laser spectrometer (QCLS). We have detected lower COS mole fractions from inland, which is likely driven by vegetation and soil uptake, and found no indications that the mud flats and salt marshes at the coast are a net sink or a net source. The nighttime COS fluxes were determined to be $-3.0 \pm 2.6$ pmol m$^{-2}$ s$^{-1}$ using the radon-tracer correlation approach. Furthermore, local sources of COS

were identified in the province of Groningen. Among these, emissions were measured at biodigesters and facilities related to organic waste processing. Biodigesters and organic waste are currently not included in emission databases of COS. However, the COS emissions have not been linked to specific process capacities or resources consumption, which currently limits



the upscaling of these newly found sources for modelling purposes. The same issues apply to agricultural soils, which could not be fully proven as a COS source or sink.

We simulated both COS and $CO_2$ concentrations at the Lutjewad station using STILT, and found that part of the observed COS enhancements can be explained by known industrial areas in Europe, such as the Ruhr area or the harbors of Antwerp and Rotterdam. Nonetheless, strong emissions during explosions occurring at ESD-SiC, a silicon carbide producer in the province of Groningen, could potentially explain large COS enhancements that were associated with easterly wind conditions. Our study demonstrates that the influence of local to regional anthropogenic sources should be considered when using COS measurements as a tracer for GPP, especially for atmospheric measurements that are close to urban areas.

**Data availability**

The data used in this work are available from https://doi.org/10.5281/zenodo.7409361. The datasets include mole fraction measurements of COS, $CO_2$, CO and $H_2O$ made in Lutjewad between 2014-2018 and the modelled results for both COS and $CO_2$.

**Funding details**

This research was supported by the ERC-advanced funding scheme (AdG 2016 Project Number: 742798, Project Acronym: COS-OCS), by the NOAA contract NA13OAR4310082, by the Dutch Research Council (NWO, grant number 184.034.015) and ICOS NL.

**Disclosure statement**

The authors declare that they have no conflict of interest.

**Author contribution**

LMJK collected the data at the Lutjewad tower. The Lutjewad tower data was analyzed and interpreted by LMJK, HAS and AS. The mobile data was collected by AZ and SvH and interpreted by AZ, SvH and HC. UK provided the particle dispersion files from the STILT model to be combined with COS databases. These were provided by JM and MK (background concentrations and anthropogenic fluxes) and LMJK (biospheric fluxes). The modelled results were interpreted by AZ, HC and SvH. The manuscript is the result of contributions by AZ, LMJK, IM and HC.

**Acknowledgement**

We thank the technical staff from the Center for Isotope research in Groningen. In particular we would like to thank Bert Kers, Marcel de Vries and Marc Bleeker for their efforts in organizing and maintaining the measurement campaigns.

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
