# Peer review of "Sources and sinks of carbonyl sulfide inferred from tower and mobile atmospheric observations"

_EGUsphere, 2023_

## Referee Comment (RC1)

Central to this study is the Stochastic Time-Inverted Lagrangian Transport (STILT) model used in combination with anthropogenic emissions inventories (Zumkehr et al., 2018) and biospheric fluxes (from the SiB4 model) to simulate the COS mole fractions at the Lutjewad (LUT) tall tower for the months of January and February 2018. Moreover, in September and October 2019, the authors carried out their own surveys of local anthropogenic sources of COS in the province where the LUT station belongs. For that purpose, in-situ measurements were made on a mobile van using a quantum cascade laser spectrometer (QCLS). These original approaches are valuable for improving our understanding of the sources and sinks of COS in The Netherlands and in Western Europe in general. However, I cannot recommend the publication of this manuscript in its present form because the study is not sufficiently set in the context of former ones. Major revision is required. The three major elements of context that have been overlooked are the following.

**Monitoring of COS over Western Europe.** Atmospheric COS has been monitored discontinuously for several years (2014-2018) at the Lutjewad tall tower (LUT) in The Netherlands (NL). The whole dataset is of high scientific value because sites where atmospheric COS has been monitored are too few in Europe. Four among the five European monitoring sites are located in Western Europe (LUT-NL (Kooijmans et al., 2016), MHD-IE (Montzka et al., 2007), GIF-FR and TRN-FR (Belviso et al., 2022a)), all gathered in a latitudinal band extending from 48°N to 53°N. Note that atmospheric COS is also monitored discontinuously in the city of Utrecht (UTR-NL) since October 2020 (Baartman et al., 2022). The following comparison of LUT, UTR, MHD and GIF datasets provides strong indication that The Netherlands are a general net source of COS during autumn and winter. The excess of atmospheric COS in The Netherlands is at least equal to 50 ppt. I recommend the authors to highlight these observations as an introduction to their finer scale approaches to sources and sinks of COS in Western Europe. Please better justify why the STILT approach has been applied only to the 2018 survey (see panel A below) when other records exist in the NL (panels B and C).

[Figure]

In section 4.1, the authors concluded that the largest excess of COS recorded at the LUT station between February 5-10 could not be ascribed to air transport from anthropogenic sources inventoried by Zumkehr et al. (2018). Only smaller enhancements measured between February 14-15 were ascribed to known European industrial areas including the Ruhr and the Antwerp-Rotterdam-Amsterdam areas. It would be very interesting to apply the STILT approach to the second large episode of COS accumulation in LUT's atmosphere dated October 2014. Moreover, because the UTR station is located closer than LUT to the potentially important Belgian-Dutch sources of anthropogenic COS, I would recommend the authors to apply the STILT approach to the UTR area too.

**COS seasonal cycle amplitudes over Western Europe.** LUT data is also used to investigate the amplitude of the seasonal variations at this site (cf. Fig. S1 copied below). In the legend of Fig. S1, the authors state that "The seasonal cycle shows a peak-to-peak amplitude of 87 ppt, which was estimated to be 96 ppt by Kooijmans et al. (2016) when no flask measurements were included." I recommend the authors to compare their observations during the period 2014-2018 with the atmospheric seasonal cycle amplitudes (SCA) assessed over MHD (Montzka et al., 2007) and GIF (Belviso et al., 2022b).

[Figure]

*Figure S1: Seasonal cycle of daytime average COS mole fractions at 60 m in Lutjewad. The data consist of in-situ measurements from August 2014 – April 2015 and January – February 2018 (circles) and flask measurements between December 2013 and February 2016 (stars). The in-situ measurements from August 2014 – April 2015 are an update of the measurements presented in Kooijmans et al. (2016). The seasonal cycle shows a peak-to-peak amplitude of 87 ppt, which was estimated to be 96 ppt by Kooijmans et al. (2016) when no flask measurements were included.*

**Seasonal Cycle Amplitude (SCA)**

[Figure]

[Figure]

COS SCA is significantly lower (about 15 ppt lower) at LUT than at MHD or GIF. What are the implications for biogenic fluxes of lower SCA in The Netherlands than elsewhere in Western Europe? How does the COS background of Fig. S1 compare with that estimated using the end point of the STILT model trajectories in the analysis domain and the derived 3D concentration fields from the Transport Model 5 – Four-Dimensional Variational model (TM5-4DVAR) inversions (Ma et al., 2021)?

**Evaluation of SiB4 simulations at LUT using observed nighttime fluxes of COS.** The authors estimated nighttime fluxes of COS based on the radon-tracer method, but, surprisingly, did not make any further use of those estimates in the manuscript. The authors could take the opportunity to compare SiB4 simulations of nighttime biogenic fluxes at LUT with field observations.

Because the STILT simulations are of central importance to the study, the way the STILT methodology is illustrated in the manuscript (cf. Fig. 2) is very disappointing. Figure 2c identifies the sources influencing Lutjewad in North-Eastern Germany 10 days before the start of the atmospheric COS survey at LUT. The associated COS enhancement is in the range 0-2 ppt/grid cell. One would conclude that the impact of the sources inventoried by Zumkehr et al. (2018) in North-Eastern Germany is estimated to be negligible to a COS enhancement that has not been quantified in the field... In fact, Fig 2c does not really identify the sources influencing Lutjewad in NE Germany because the anthropogenic and biogenic contributions are not separated from each other. Moreover, the color scale adopted in Fig. 2b does not allow at all localizing the direct COS sources inventoried by Zumkehr et al. (2018). I would use a log scale in the range 1 to 1000 pmol m$^{-2}$ s$^{-1}$. Figures 2b and 2c are misleading and should be redrawn. A date belonging to period 4 could be chosen to better illustrate the enhancements attributed to industry in the Ruhr area. Larger panels are required. It would be also interesting to document the largest excess of COS recorded at the LUT station between February 5-10 (period 3) the one that could not be ascribed to air transport from anthropogenic sources inventoried by Zumkehr et al. (2018).

Other methodological aspects to be clarified are the following:

-100 particles released for 10 days back in time: isn't it a too small number of particles?

-Is the horizontal resolution of the ECMWF-IFS database of 0.1°x0.1° or coarser?

-At what time are the particles released to the atmosphere?

Figure 7. Again, I don't understand the reason why the authors provided modelled COS concentrations when observations are not available (e.g., Fig. 7, right column, red curves, dates before 01-18-2018 17:00 and after 02-19-2018 8:00:00). The consequences are that the difference between measurements (black curve) and modelled values (red curves) are poorly visible. Please redraw Figure 7 accordingly. As an alternative, the contributions of background, background + biogenic fluxes, background + biogenic fluxes + direct anthropogenic emissions, background + biogenic fluxes + direct & indirect anthropogenic emissions could be displayed on the same plot. Data displayed in Fig. 7 and Fig. 9 could be combined by plotting background + biogenic fluxes + direct & indirect anthropogenic emissions + local sources identified from mobile flask and in-situ measurements.

I also question the interest of Figure 3, where the deviation of mole fractions of COS from their seasonal cycle in Lutjewad is compared, because the COS background at 60 m set from data gathered

in Fig. S1 is not well constrained for the months of January and February. I would rather suggest the use of cluster analysis applied to HYSPLIT back trajectories calculated every 3 h at the LUT site during the months of January and February 2018.

Other comments of less importance are listed below:

-Title: Sources and sinks of carbonyl sulfide inferred from tower and mobile atmospheric observations in The Netherlands

-page 2, line 36: remove "on average"

-page 3, line 5: NOAA data can be visualized on-line at https://gml.noaa.gov/dv/iadv/

-page 3, line 9: …were analyzed by gas chromatography and mass spectrometry.

-page 3, line 18: Moreover, this instrument enabled the collection of…

-page 6, line 8 and Table 2: no overview of the average precision is given in Table 2. Remove Table 2.

-page 12, line 12: CO molar fractions are not displayed in Fig. 5.

Page 13: this very descriptive paragraph should be rewritten in order to better identify the data in Fig. 5 and Fig. 6 to which the authors refer to. A letter should be attributed to each panel to guide the reader.

Page 16: Please provide an illustration of how the COS fluxes were calculated with in-situ measurements collected at ground level. Is it realistic to use a Gaussian dispersion model when the vertical distribution of COS remains unknown? Was a 3D sonic anemometer coupled with the QCLS?

Page 19, line 6: Do you mean that rapeseed is grown in the Groningen province in spring and that soils are fertilized in winter with rapeseed byproducts?

Page 19, line 31: Are you aware of any explosions at ESD-Sic in October 2014 when atmospheric COS levels at LUT were over 500 ppt?

Last remark.

I would like to inform you of the existence of a manuscript entitled "The Z-2018 emissions inventory of COS in Europe: a semiquantitative multi-data-streams evaluation", authored by I. Pison, J.-E. Petit, A. Berchet, M. Remaud, L. Simon, M. Ramonet, M. Delmotte, V. Kazan, C. Yver-Kwok, M. Lopez and myself (S. Belviso), in press in Atmospheric Environment. I will be keen to share a preprint with you upon request. Chapter 3.3 provides examples of cluster analysis of winter COS measurements and back trajectories. One event is dated February 2018.

References cited:

Baartman, S.L., Kroll, M.C., Röckmann, T., Hattori, S., Kamesaki, K., Yoshida, N., Popa, M.E., 2022. A GC-IRMS method for measuring sulfur isotope ratios of carbonyl sulfide from small air samples. Open Research Europe 2022, 1:105. https://doi.org/10.12688/openreseurope.13875-2

Belviso, S., Abadie, C., Montagne, D., Hadjar, D., Tropée, D., Vialettes, L., Kazan, V., Delmotte, M., Maignan, F., Remaud, M., Ramonet, M., Lopez, M., Yver-Kwok, C., Ciais, P., 2022a. Carbonyl sulfide

(COS) emissions in two agroecosystems in central France. PLoS ONE 17(12): e0278584. https://doi.org/10.1371/journal.pone.0278584

Belviso, S., Remaud, M., Abadie, C., Maignan, F., Ramonet, M., Peylin, P., 2022b. Ongoing Decline in the Atmospheric COS Seasonal Cycle Amplitude over Western Europe: Implications for Surface Fluxes. Atmosphere 13, 812. https://doi.org/10.3390/atmos13050812

Kooijmans, L. M. J., Uitslag, N. A. M., Zahniser, M. S., Nelson, D. D., Montzka, S. A., & Chen, H. (2016). Continuous and high-precision atmospheric concentration measurements of COS, CO2, CO and H2O using a quantum cascade laser spectrometer (QCLS). Atmospheric Measurement Techniques, 9(11), 5293–5314. https://doi.org/10.5194/amt-9-5293-2016

Montzka, S.A., Calvert, P., Hall, B.D., Elkins, J.W., Conway, T.J., Tans, P.P., Sweeney, C., 2007. On the global distribution, seasonality, and budget of atmospheric carbonyl sulfide (COS) and some similarities to CO2. J. Geophys. Res. 112, D09302. https://doi.org/10.1029/2006JD007665

---

## Author Comment (AC1)

**Comment on egusphere-2023-208**
https://doi.org/10.5194/egusphere-2023-208
Saveur Belviso

Referee comment by Saveur Belviso (sauveur.belviso@lsce.ipsl.fr) on "Sources and sinks of carbonyl sulfide inferred from tower and mobile atmospheric observations" by Zanchetta et al., Biogeosciences Discussion, https://doi.org/10.5194/egusphere-2023-208, 2023.

Central to this study is the Stochastic Time-Inverted Lagrangian Transport (STILT) model used in combination with anthropogenic emissions inventories (Zumkehr et al., 2018) and biospheric fluxes (from the SiB4 model) to simulate the COS mole fractions at the Lutjewad (LUT) tall tower for the months of January and February 2018. Moreover, in September and October 2019, the authors carried out their own surveys of local anthropogenic sources of COS in the province where the LUT station belongs. For that purpose, in-situ measurements were made on a mobile van using a quantum cascade laser spectrometer (QCLS). These original approaches are valuable for improving our understanding of the sources and sinks of COS in The Netherlands and in Western Europe in general. However, I cannot recommend the publication of this manuscript in its present form because the study is not sufficiently set in the context of former ones. Major revision is required. The three major elements of context that have been overlooked are the following.

**Monitoring of COS over Western Europe.** Atmospheric COS has been monitored discontinuously for several years (2014-2018) at the Lutjewad tall tower (LUT) in The Netherlands (NL). The whole dataset is of high scientific value because sites where atmospheric COS has been monitored are too few in Europe. Four among the five European monitoring sites are located in Western Europe (LUT- NL (Kooijmans et al., 2016), MHD-IE (Montzka et al., 2007), GIF-FR and TRN-FR (Belviso et al., 2022a)), all gathered in a latitudinal band extending from 48°N to 53°N. Note that atmospheric COS is also monitored discontinuously in the city of Utrecht (UTR-NL) since October 2020 (Baartman et al., 2022). The following comparison of LUT, UTR, MHD and GIF datasets provides strong indication that The Netherlands are a general net source of COS during autumn and winter. The excess of atmospheric COS in The Netherlands is at least equal to 50 ppt. I recommend the authors to highlight these observations as an introduction to their finer scale approaches to sources and sinks of COS in Western Europe. Please better justify why the STILT approach has been applied only to the 2018 survey (see panel A below) when other records exist in the NL (panels B and C).

***Answer***: *the authors thank the referee for this useful insight that had been overlooked. These observations stress even more the importance of a further investigation of local sources at a regional level for the Netherlands. The following paragraph has been added at page 3, lines 21-29 in Section 1 (Introduction):*

*Tropospheric COS molar fraction is only monitored in a few sites in Europe. Among these, four monitoring sites are located in Western Europe, within 48°N and 53°N: Mace Head, Ireland (Montzka et al., 2007), Gif-sur-Yvette and Trainou, France (Belviso et al., 2022) and Lutjewad, the Netherlands (Kooijmans et al., 2016). Moreover, COS has been recently monitored discontinuously in Utrecht, the Netherlands (Baartman et al., 2022). The observations in these studies show higher autumn and winter COS molar fractions in the Netherlands than those at Gif-sur-Yvette and Trainou, France. This calls for a more thorough investigation of possible local sources in the Netherlands at a local and regional scale.*

*Concerning the STILT application, the choice was led mainly by the unusuality of the observations in 2018 and we mostly aimed to disentangle the influence of local and regional sources on these observations. In particular, given the extremely high molar fractions*

*registered in 2018 and the initial suspects on ploughing activities as a COS source, the authors chose to focus on this particular period alone.*

*The sentence starting at page 7, line 43 was modified as follows:*

*The simulation covers the period from January to February 2018, given the availability for both observations and models data and unusually high COS molar fractions (see Section 3.3) for this period. This analysis aims to disentangle the influence of local and regional sources on these observations.*

[Figure]

In section 4.1, the authors concluded that the largest excess of COS recorded at the LUT station between February 5-10 could not be ascribed to air transport from anthropogenic sources inventoried by Zumkehr et al. (2018). Only smaller enhancements measured between February 14-15 were ascribed to known European industrial areas including the Ruhr and the Antwerp-Rotterdam-Amsterdam areas. It would be very interesting to apply the STILT approach to the second large episode of COS accumulation in LUT's atmosphere dated October 2014. Moreover, because the UTR station is located closer than LUT to the potentially important Belgian-Dutch sources of anthropogenic COS, I would recommend the authors to apply the STILT approach to the UTR area too.

***Answer***: *the authors thank the referee for this remark and agree it could be very interesting to apply a thorough analysis to the UTR area as well as to other periods when continuous measurements were realized in LUT. However, the authors believe this addition would better fit within a further study aiming to compare regional and/or continental influences on COS monitoring sites and would therefore be beyond the purpose of the study presented in this paper. A combined effort to assess areas that bias COS measurements in different monitoring sites would surely provide a relevant addition to the understanding of COS sources.*

**COS seasonal cycle amplitudes over Western Europe**. LUT data is also used to investigate the amplitude of the seasonal variations at this site (cf. Fig. S1 copied below). In the legend of Fig. S1, the

authors state that "The seasonal cycle shows a peak-to-peak amplitude of 87 ppt, which was estimated to be 96 ppt by Kooijmans et al. (2016) when no flask measurements were included." I recommend the authors to compare their observations during the period 2014-2018 with the atmospheric seasonal cycle amplitudes (SCA) assessed over MHD (Montzka et al., 2007) and GIF (Belviso et al., 2022b).

[Figure]

Figure S1: Seasonal cycle of daytime average COS mole fractions at 60 m in Lutjewad. The data consist of in-situ measurements from August 2014 – April 2015 and January – February 2018 (circles) and flask measurements between December 2013 and February 2016 (stars). The in-situ measurements from August 2014 – April 2015 are an update of the measurements presented in Kooijmans et al. (2016). The seasonal cycle shows a peak-to-peak amplitude of 87 ppt, which was estimated to be 96 ppt by Kooijmans et al. (2016) when no flask measurements were included.

[Figure]

COS SCA is significantly lower (about 15 ppt lower) at LUT than at MHD or GIF. What are the implications for biogenic fluxes of lower SCA in The Netherlands than elsewhere in Western Europe? How does the COS background of Fig. S1 compare with that estimated using the end point of the STILT model trajectories in the analysis domain and the derived 3D concentration fields from the Transport Model 5 – Four-Dimensional Variational model (TM5-4DVAR) inversions (Ma et al., 2021)?

**Answer**: *the differences in COS SCA's have already been mentioned in Kooijmans et al. (2016), where the COS SCA was analyzed for LUT in 2014-2015. This is also the reason why this analysis, which has been realized by adding in-situ observations to the same dataset, was reported as supplementary information. The SCA's have been depicted in Figure 13 of Kooijmans et al. (2016), which has been copied below.*

[Figure]

**Figure 13.** COS seasonal cycle of four sites: Wisconsin, USA (LEF); Mauna Loa, USA (MLO); Mace Head, Ireland (MHD); and Lutjewad, the Netherlands (LUT); the data of the latter site are presented in this study. COS mole fractions for the LEF, MLO and MHD sites were measured from flask samples at a GC-MS by NOAA/ESRL (Montzka et al., 2007). The NOAA/ESRL data are shown as flask pair means from individual sampling events. All NOAA measurements are plotted as function of time of the year and cover a period between 2000 and 2015 for LEF, MLO and MHD. In situ COS measurements with the QCLS at the Lutjewad site during 2014–2015 are shown as daily averages (black). A two-harmonic seasonal cycle is fit through the data.

*Citing Kooijmans et al. (2016):*

*"In Fig. 13 we compare COS mole fractions from Lutjewad with that from three other sites as measured from flask samples with a GC-MS by NOAA/ESRL Montzka et al. (2007). The flask samples cover data between 2000 and 2015 for Wisconsin, USA (LEF), and Mauna Loa, USA (MLO), and between 2001 and 2015 for Mace Head, Ireland (MHD). These data are an update of those presented in Montzka et al. (2007) (data available at ftp://ftp.cmdl.noaa.gov/hats/carbonyl_sulfide/). The flask measurements in Fig. 13 are plotted as function of time of the year. The high-altitude MLO site is less directly influenced by terrestrial ecosystems and therefore shows only small seasonal variation, in contrast to the LEF site, which is largely influenced by (forested) continental air. The Lutjewad COS mole fractions are most consistent with measurements from MHD, which can be expected since both stations are coastal sites and are located at similar latitudes. The seasonal amplitude of COS at MHD and Lutjewad is in between that of LEF and MLO, most likely because both sites are not solely influenced by marine or continental air but by both types of air masses. The COS mole fraction has a minimum in September and October and is a few weeks later than the minimum of the $CO_2$ mole fraction. Montzka et al. (2007) and Blonquist et al. (2011) also observed a COS minimum later than that of $CO_2$. They reasoned that this difference is due to the fact that at the end of the growing season COS mole fractions keep decreasing due to vegetative uptake without at the same time having a source of COS, whereas during this time of year respiration is beginning to offset assimilation in determining the ambient $CO_2$ mole fractions".*

*It is interesting to notice that according to Kooijmans et al. (2016) and contrasting the referee's comment, MHD and LUT show very close SCA's, with Kooijmans et al. concluding that the seasonal cycle features are probably determined by the monitoring site location.*

*Concerning the agreement between TM5-4DVAR background obtained by boundary conditions and its comparison with COS SCA presented in Figure S1, the investigation is currently limited by data availability. Only particles dispersion files for January and February 2018 were made available, therefore it is only possible to compare the COS SCA of the first 59 days of the year (see Figure A below). Generally, the available modelled data falls within the range of observations, but of course this cannot*

*be considered a thorough analysis. It would be interesting to include this interesting insight in future works, possibly within a comprehensive intercomparison of different measuring stations.*

[Figure]

*Figure A: comparison of seasonal cycles of COS measured in Lutjewad and retrieved from TM5-4DVAR boundary conditions for January and February 2018.*

**Evaluation of SiB4 simulations at LUT using observed nighttime fluxes of COS**. The authors estimated nighttime fluxes of COS based on the radon-tracer method, but, surprisingly, did not make any further use of those estimates in the manuscript. The authors could take the opportunity to compare SiB4 simulations of nighttime biogenic fluxes at LUT with field observations.

Because the STILT simulations are of central importance to the study, the way the STILT methodology is illustrated in the manuscript (cf. Fig. 2) is very disappointing. Figure 2c identifies the sources influencing Lutjewad in North-Eastern Germany 10 days before the start of the atmospheric COS survey at LUT. The associated COS enhancement is in the range 0-2 ppt/grid cell. One would conclude that the impact of the sources inventoried by Zumkehr et al. (2018) in North-Eastern Germany is estimated to be negligible to a COS enhancement that has not been quantified in the field... In fact, Fig 2c does not really identify the sources influencing Lutjewad in NE Germany because the anthropogenic and biogenic contributions are not separated from each other. Moreover, the color scale adopted in Fig. 2b does not allow at all localizing the direct COS sources inventoried by Zumkehr et al. (2018). I would use a log scale in the range 1 to 1000 pmol m$^{-2}$ s$^{-1}$. Figures 2b and 2c are misleading and should be redrawn. A date belonging to period 4 could be chosen to better illustrate the enhancements attributed to industry in the Ruhr area. Larger panels are required. It would be also interesting to document the largest excess of COS recorded at the LUT station between February 5-10 (period 3) the one that could not be ascribed to air transport from anthropogenic sources inventoried by Zumkehr et al. (2018).

**Answer**: *the authors agree with the referee's comment. Unfortunately, SiB4 data for Lutjewad were requested only for January and February 2018. The average COS nighttime flux was estimated at Lutjewad coordinates over these two months and resulted to be -2.1 ± 0.2 pmol/(m²\*s). On page 10, Lines 22-24, the following sentence was added:*

*The average SiB4 COS nighttime (9PM – 6AM) flux was retrieved for Lutjewad (53.4°N, 6.3°E) for January and February 2018 and was estimated to be -2.1 ± 0.2 pmol m$^{-2}$ s$^{-1}$.*

On the other hand, the problem for Figure 2c was not the missed separation between biospheric and anthropogenic fluxes, but a missing "log" in the reported unit of measure. We have chosen 15 February 2018 of period 4, as an example, to illustrate the COS enhancements attributed to industries in the Ruhr area. The figures have been redrawn with larger panels and a log scale as suggested, adopting restricted color scales to help the visualization and the caption has been modified as reported below:

[Figure]

*Figure 2: reported in logarithmic scales: panels (a) and (b) show the localized COS and CS$_2$ sources according to Zumkehr et al. (2018), (c) shows an example of localized footprint values resulting from the STILT model simulations, summed over 10 days before the starting timestep (15/02/2018, 09:00), (d) the modelled enhancement resulting from the product of footprint and fluxes (see Section 2.4), identifying the sources influencing Lutjewad in the Ruhr area (the ranges of these scales were adjusted for clarity purposes).*

Other methodological aspects to be clarified are the following:
-100 particles released for 10 days back in time: isn't it a too small number of particles?

***Answer****: the authors believe the number of released particles for these simulations is sufficient, since it is in line with other studies related to STILT applications (see, for example, Galkowski, 2015; Gerbig et al., 2003; Maier et al., 2022; Thilakan et al., 2022; Van Der Woude et al., 2023). Maier et al. (2022), in particular, report a case-sensitivity analysis between 100 particles over a 3-days dispersion against*

*500 particles over a 10-days dispersion, which resulted only in minor differences. For this study, the 10-days-long dispersion was necessary given the estimated lifetime of CS₂.*

-Is the horizontal resolution of the ECMWF-IFS database of 0.1°x0.1° or coarser?

***Answer****: the ECMWF-IFS resolution is 0.25°x0.25°. Page 8, line 5 was modified to include this information as follows:*

*…"driven by ECMWF-IFS operational analysis at a 0.25°x0.25° resolution"…*

-At what time are the particles released to the atmosphere?

***Answer****: the particles are released to the atmosphere at the time when the observations were made, and the particles are transported backward in time based on 3-hourly wind fields covering January and February 2018.*

*Page 8, lines 11-17 were modified as follows:*

*"The STILT model establishes the link between the emissions in the upwind influencing area and the measurements at a defined location and time. This is realized by releasing particles to the atmosphere that are driven by meteorological winds and transported backward in time to determine the origin of air parcels influencing the measurements. Each simulation run releases 100 particles from the Lutjewad station, at a height of 60 m. The transport of these particles is reconstructed within the selected domain (latitude 34.0°N-73.5°N, longitude 20.0°W-45.5°E, to cover Europe), in 3-hours timesteps over 10 days back in time."*

Figure 7. Again, I don't understand the reason why the authors provided modelled COS concentrations when observations are not available (e.g., Fig. 7, right column, red curves, dates before 01-18-2018 17:00 and after 02-19-2018 8:00:00). The consequences are that the difference between measurements (black curve) and modelled values (red curves) are poorly visible. Please redraw Figure 7 accordingly. As an alternative, the contributions of background, background + biogenic fluxes, background + biogenic fluxes + direct anthropogenic emissions, background + biogenic fluxes + direct & indirect anthropogenic emissions could be displayed on the same plot. Data displayed in Fig. 7 and Fig. 9 could be combined by plotting background + biogenic fluxes + direct & indirect anthropogenic emissions + local sources identified from mobile flask and in-situ measurements.

***Answer****: the authors agree with the referee and would prefer to keep these figures separated given the different purpose of the two (e.g. showing contributions of different parameters on one and of the newly introduced sources on the other). Therefore, Figure 7 has been redrawn with a shorter x-axis, comprising only the period when measurements are available.*

I also question the interest of Figure 3, where the deviation of mole fractions of COS from their seasonal cycle in Lutjewad is compared, because the COS background at 60 m set from data gathered in Fig. S1 is not well constrained for the months of January and February. I would rather suggest the use of cluster analysis applied to HYSPLIT back trajectories calculated every 3 h at the LUT site during the months of January and February 2018.

***Answer****: Figure 3 reports the usual relationship between observed trends in COS, CO₂ and CO molar fraction enhancements or depletions and wind directions, compared to the average measured seasonal*

*cycle. The intention of both Section 3.1 and Section 3.2 is to show the results which set the context for the measurements described in the following paragraphs. However, the authors recognize Section 3.1 was rather self-standing and not well contextualized within this study. Therefore, it has been moved together with Figure 3 and Section 4.2 to the supplementary materials. The title of the new Section S1 was modified as "Observed deviations from seasonal cycles by wind directions during stationary measurements". Figure 3 has now become Figure S1.*

Other comments of less importance are listed below:

-Title: Sources and sinks of carbonyl sulfide inferred from tower and mobile atmospheric observations in the Netherlands

***Answer****: the authors agree with the referee's suggestion. The title has been modified accordingly.*

-page 2, line 36: remove "on average"

***Answer****: done.*

-page 3, line 5: NOAA data can be visualized on-line at https://gml.noaa.gov/dv/iadv/

***Answer****: on page 3, line 8, it was added "This dataset is still being updated and can be visualized online (NOAA, 2023)."*

-page 3, line 9: ...were analyzed by gas chromatography and mass spectrometry.

***Answer****: corrected.*

-page 3, line 18: Moreover, this instrument enabled the collection of...

***Answer****: corrected.*

-page 6, line 8 and Table 2: no overview of the average precision is given in Table 2. Remove Table 2.

***Answer****: page 6, line 8 was modified as "the average precision for the 2014-2015 period was 5.3 ppt". However, the authors prefer to keep Table 2 since it reports also the different measurement heights and times in the measurement periods.*

-page 12, line 12: CO molar fractions are not displayed in Fig. 5.

***Answer****: the sentence has been modified: "...$CO_2$ and CO (this latter not shown in Figure 4) molar fractions…"*

Page 13: this very descriptive paragraph should be rewritten in order to better identify the data in Fig. 5 and Fig. 6 to which the authors refer to. A letter should be attributed to each panel to guide the reader.

***Answer****: the authors included letters in the panels and references to Figure 5 and Figure 6 throughout the paragraph. However, the authors would prefer to keep the paragraph in its current form since it includes some of the key findings of this study – namely, the mismatch between modelled results and measurements in Period 3 – which could be better described with a thorough description and comparison against the other selected periods of interest.*

Page 16: Please provide an illustration of how the COS fluxes were calculated with in-situ measurements collected at ground level. Is it realistic to use a Gaussian dispersion model when the vertical distribution of COS remains unknown? Was a 3D sonic anemometer coupled with the QCLS?

**Answer**: *an example of measurements is provided in Figure B below (location: SuikerUnie, 53°12'N, 6°30'E). The authors acknowledge that major simplifications were made to apply the Gaussian dispersion model for these estimations. The peaks measured during the mobile sampling campaign closely followed a Gaussian shape. Unfortunately, there were no 3D-sonic measurements coupled with this sampling campaign. Furthermore, in most cases the exact location of the emission sources remained unknown (e.g., it was not possible to see any industrial chimney, or there was no visible plume). Therefore, it was necessary to assume that emissions occurred at the sampling height. The emissions were reconstructed using the parametrization of $\sigma_y$ and $\sigma_z$ defined after Pasquill-Gifford stability classes (Csanady, 1973), estimated after a Monte Carlo simulation based on distance from the source and wind speed (obtained from approximated estimates from Google Maps and weather data). This, in fact, results in fluxes uncertainties that range between 44% and 92% of the estimated flux means. The authors would like to stress that the Gaussian plume modelling was applied to obtain some rough estimate; the used approach is not considered a reliable representation of reality, but rather a tool to get the best estimate possible given the available data.*

[Figure]

*Figure B: COS in-situ measurements at SuikerUnie, now CosunBeet Company, 53°12'N, 6°30'E (Hoogkerk, Groningen, NL).*

Page 19, line 6: Do you mean that rapeseed is grown in the Groningen province in spring and that soils are fertilized in winter with rapeseed byproducts?

***Answer**: indeed, rapeseed was grown in some fields of the Groningen province in spring and the byproducts were occasionally used as fertilizers. The sentence on page 17, line 25 was modified as follows: …"and, knowing that rapeseed was grown in some fields in the province of Groningen, it is still possible that a fertilizer based on rapeseed byproducts (Belviso et al., 2022) or the soil act as a COS source occasionally"…*

Page 19, line 31: Are you aware of any explosions at ESD-Sic in October 2014 when atmospheric COS levels at LUT were over 500 ppt?

***Answer**: unfortunately, it was not possible to obtain detailed records of all the explosions occurred at ESD-SiC preceding 2018. An interrogation presented at the Second Chamber of the Netherlands (Lacin, 2019) (Dutch only) reports 148 explosions between 2014 and 2017. A TNO investigation on the composition of soot in ESD-SiC explosions (Tromp & Duyzer, 2019) (Dutch only), reports the explosions per year between 2014-2018 to be 50, 35, 29, 34 and 30, respectively. Moreover, a local newspaper*

reported the villages of Meedhuizen and Tjuchem, situated West of ESD-SiC, to be covered in soot following an explosion at ESD-SiC on Friday, January 23rd 2015 (RTVNoord, 2015). As shown in Figure C, on such date COS enhancements were measured in Lutjewad for about 3 hours. Unfortunately, particle dispersion files were not (made) available for 2015 and therefore it was not possible to perform a thorough analysis of air transport for this period. However, the gathered information suggests that, at least for January 2015, ESD-SiC may have already been the cause of the measured enhancements.

[Figure]

*Figure C: measured COS in Lutjewad between December 2014 and January 2015.*

Last remark.

I would like to inform you of the existence of a manuscript entitled "The Z-2018 emissions inventory of COS in Europe: a semiquantitative multi-data-streams evaluation", authored by I. Pison, J.-E. Petit, A. Berchet, M. Remaud, L. Simon, M. Ramonet, M. Delmotte, V. Kazan, C. Yver-Kwok, M. Lopez and myself (S. Belviso), in press in Atmospheric Environment. I will be keen to share a preprint with you upon request. Chapter 3.3 provides examples of cluster analysis of winter COS measurements and back trajectories. One event is dated February 2018.

***The authors would like to thank the referee for the useful comments, which helped to get a more comprehensive overview for this study and to contextualize it within the existing observational network.*** *A last sentence was added at the end of the manuscript (page 20, lines 17-22): "Our study demonstrates that the influence of local to regional anthropogenic sources should be considered when using COS measurements as a tracer for GPP, especially for atmospheric measurements that are close to urban areas. This approach, combining COS stationary measurements, mobile measurements and models, could be applied in other existing measurement locations. It could allow a broader assessment of local anthropogenic influences, to prevent biases in COS budget and seasonality estimates.".*

References cited:

Baartman, S.L., Kroll, M.C., Röckmann, T., Hattori, S., Kamesaki, K., Yoshida, N., Popa, M.E., 2022. A GC-IRMS method for measuring sulfur isotope ratios of carbonyl sulfide from small air samples. Open Research Europe 2022, 1:105. https://doi.org/10.12688/openreseurope.13875-2

Belviso, S., Abadie, C., Montagne, D., Hadjar, D., Tropée, D., Vialettes, L., Kazan, V., Delmotte, M., Maignan, F., Remaud, M., Ramonet, M., Lopez, M., Yver-Kwok, C., Ciais, P., 2022a. Carbonyl sulfide (COS) emissions in two agroecosystems in central France. PLoS ONE 17(12): e0278584. https://doi.org/10.1371/journal.pone.0278584

Belviso, S., Remaud, M., Abadie, C., Maignan, F., Ramonet, M., Peylin, P., 2022b. Ongoing Decline in the Atmospheric COS Seasonal Cycle Amplitude over Western Europe: Implications for Surface Fluxes. Atmosphere 13, 812. https://doi.org/10.3390/atmos13050812

Kooijmans, L. M. J., Uitslag, N. A. M., Zahniser, M. S., Nelson, D. D., Montzka, S. A., & Chen, H. (2016). Continuous and high-precision atmospheric concentration measurements of COS, CO2, CO and H2O using a quantum cascade laser spectrometer (QCLS). Atmospheric Measurement Techniques, 9(11), 5293–5314. https://doi.org/10.5194/amt-9-5293-2016

Montzka, S.A., Calvert, P., Hall, B.D., Elkins, J.W., Conway, T.J., Tans, P.P., Sweeney, C., 2007. On the global distribution, seasonality, and budget of atmospheric carbonyl sulfide (COS) and some similarities to CO2. J. Geophys. Res. 112, D09302. https://doi.org/10.1029/2006JD007665

_References cited in the response_:

Csanady, G. T. (1973). _Turbulent Diffusion in the Environment_. Springer Netherlands.

https://doi.org/10.1007/978-94-010-2527-0

Galkowski, M. (2015). _Temporal and Spatial Variability of Nitrous Oxide in the Atmosphere_

_over Małopolska Region: Determination of Loads and Emissions_.

https://doi.org/10.13140/RG.2.2.13520.10248

Gerbig, C., Lin, J. C., Wofsy, S. C., Daube, B. C., Andrews, A. E., Stephens, B. B., Bakwin, P. S.,

& Grainger, C. A. (2003). Toward constraining regional-scale fluxes of $CO_2$ with

atmospheric observations over a continent: 2. Analysis of COBRA data using a

receptor-oriented framework: TOWARD REGIONAL-SCALE FLUXES OF $CO_2$, 2. _Journal_

_of Geophysical Research: Atmospheres_, _108_(D24), n/a-n/a.

https://doi.org/10.1029/2003JD003770

Lacin, Ç. (2019). *Vragen gesteld door de leden der Kamer, met de daarop door de regering gegeven antwoorden*. ah-tk-20192020-1334. https://zoek.officielebekendmakingen.nl/ah-tk-20192020-1334.pdf

Maier, F., Gerbig, C., Levin, I., Super, I., Marshall, J., & Hammer, S. (2022). Effects of point source emission heights in WRF–STILT: A step towards exploiting nocturnal observations in models. *Geoscientific Model Development*, *15*(13), 5391–5406. https://doi.org/10.5194/gmd-15-5391-2022

RTVNoord. (2015, January 23). *Meedhuizen en Tjuchem kleuren zwart van de roet*. https://www.rtvnoord.nl/nieuws/144256/meedhuizen-en-tjuchem-kleuren-zwart-van-de-roet

Thilakan, V., Pillai, D., Gerbig, C., Galkowski, M., Ravi, A., & Anna Mathew, T. (2022). Towards monitoring the $CO_2$ source–sink distribution over India via inverse modelling: Quantifying the fine-scale spatiotemporal variability in the atmospheric $CO_2$ mole fraction. *Atmospheric Chemistry and Physics*, *22*(23), 15287–15312. https://doi.org/10.5194/acp-22-15287-2022

Tromp, P., & Duyzer, J. (2019). *Resultaten onderzoek naar het verspreidingsgebied van gedeponeerd stof als gevolg van blazers* (100320545 versie 2). TNO. https://www.provinciegroningen.nl/fileadmin/user_upload/Documenten/Downloads/Downloads_2019/Onderzoeksrapport_TNO_naar_SiC-vezels_in_sneeuw_na_blazer_24-januari_11-april-2019_herziene-versie.pdf

Van Der Woude, A. M., De Kok, R., Smith, N., Luijkx, I. T., Botía, S., Karstens, U., Kooijmans, L. M. J., Koren, G., Meijer, H. A. J., Steeneveld, G.-J., Storm, I., Super, I., Scheeren, H. A., Vermeulen, A., & Peters, W. (2023). Near-real-time $CO_2$ fluxes from CarbonTracker

Europe for high-resolution atmospheric modeling. *Earth System Science Data*, *15*(2), 579–605. https://doi.org/10.5194/essd-15-579-2023

---

## Author Comment (AC2)

**Comment on egusphere-2023-208**

https://doi.org/10.5194/egusphere-2023-208

Mary Whelan

Referee comment by Mary Whelan (mary.whelan@gmail.com) on "Sources and sinks of carbonyl sulfide inferred from tower and mobile atmospheric observations" by Zanchetta et al., Biogeosciences Discussion, https://doi.org/10.5194/egusphere-2023-208, 2023.

Dear Authors,

An understanding of atmospheric OCS sources and sinks enable the ability to ascertain plant functioning on an integrated, regional scale inaccessible to other methods. This manuscript presents an effort in untangling OCS gross fluxes over a specific region. Some additional analysis and editing are needed to realize the potential of the study. Below I have some major questions followed by a few minor ones.

*The authors would like to thank the referee for the generally positive comments and for the insightful remarks and questions.*

*The responses will be organized question-by-question in paragraphs formatted similarly to the present one. Major modifications in the preprint will be presented as underlined text together with their respective page and line numbers.*

For the first investigation that Kooijmans et al published in 2016, there is an entire year of calibrated data, but the additional data presented here is at a single tower measurement height for 2 months without a calibration cylinder? Is there something missing in the description in the text?

***Answer***: *the measurements in January and February 2018 were only performed at 60m height due to necessary maintenance work on the 7m and 40m height sampling lines. Moreover, as the referee correctly underlines, no target cylinders were measured in that period. However, as stated on page 5, line 16: "A reference cylinder was measured every half hour to correct for instrument drift and to calibrate the measurements to the common scales". Therefore, there was no target gas to independently assess the stability of the measurements, but it was made sure that the measurements were corrected for drift and that they all fell within a common measurement scale.*

In this study, the tower footprints over time were calculated by STILT and the concentrations of the tower were calculated based on assumed fluxes at the surface. Attribution was estimated on page 13 based on footprints during periods of trace gas enhancement.

If we want to do something truly powerful with this data, we can take the known flux estimates as priors and generate new maps of surface fluxes based on observed concentrations at the tower (averaged over afternoons where nighttime inversions have already been dispensed with. Calculating footprints when the PBL is on the move, e.g. at midnight, is error-prone.) This atmospheric inversion would give you a stronger, data-based hint about where the missing sources of the region are and requires no further field measurements.

That said, the uncertainty introduced by using the STILT model is not sufficiently addressed. Derek Mallia at the University of Utah writes articulately about the STILT model and it's application to regional fluxes. The recent update to the STILT model – which version did you use? – makes the analysis more user friendly than previous versions. There has also been work done by Anna Michalak's group in analyzing uncertainties in this type of analysis.

**Answer**: *the authors agree that an atmospheric inversion approach in combination with tower measurements could provide a powerful tool to locate missing sources at a regional level. The authors also believe it could be interesting to work on a comparative study, to investigate how well local or regional sources inferred from an atmospheric inversion and measured ones would agree. However, the application of this method may be over the scope of this study, which rather aims to introduce a measurements-based technique to identify local sources that could bias tower measurements. A thorough evaluation based on forward model runs and a detailed quantification are prerequisites before applying an inversion. Also, the conclusions from an inversion would be stronger if measurements at several stations rather than at one station could be used to cover a region, which will actually be investigated in our followup study.*

*The STILT model was based on ECMWF-IFS cycle 47r1 (see [https://confluence.ecmwf.int/display/FCST/Implementation+of+IFS+Cycle+47r1](https://confluence.ecmwf.int/display/FCST/Implementation+of+IFS+Cycle+47r1)). The authors acknowledge that uncertainties in transport were not addressed quantitatively and that the inversion modelling approach was not evaluated thoroughly. However, this footprint analysis was performed to identify areas of influence for Lutjewad measurements. Secondly, it aimed to give an order-of-magnitude estimate of the possible impact of newly discovered sources on stationary measurements. Concerning the first application, the authors did not quantify uncertainties in particles transport and footprint locations. Nonetheless, both the extension of possible areas of influence and the spread within anthropogenic COS fluxes magnitude (0-500.4 pmol/(m$^2$*s) for direct COS, 0-1421.5 pmol/(m$^2$*s) for CS$_2$) provided information that the authors considered to be reliable enough for the identification of influential zones (an example is provided in Figure A below, which has also been included in the main text as a replacement of Figure 2). With regard to the impact of newly found sources on the measurements, instead, the uncertainty on fluxes was estimated by performing a Monte Carlo simulation (see Section 3.4 in the preprint), but there was no quantitative assessment of uncertainties for the footprint output. The only sensitivity analysis was applied to CS$_2$ lifetimes (3 to 10 days) and led to minor differences. The authors acknowledge that this approach is approximated and that*

*major uncertainties may arise in particular from the Planetary Boundary Layer height and convective vertical transport. However, these results were considered sufficient for the desired order-of-magnitude estimate and to prove that, on specific dates, local sources could have actually biased stationary measurements in Lutjewad significantly.*

[Figure]

*Figure A: (a) direct and (b) indirect anthropogenic COS fluxes, (c) footprints calculated by the STILT model (d) localized effects on Lutjewad measurements obtained by combining (a), (b) and (c) – see main text in the preprint (Sections 2 and 3) for a thorough description.*

The analysis in 3.1 may belong in the supplement with Figure S1. It is a look at wind direction and deviation from a calculated seasonal average. Using the flux-gradient method or approach, a flux estimate could be made based on concentrations measured at two different heights (along with high frequency wind and temperature data).  However, the conclusion of the analysis here is unsatisfying – we are no closer to knowing the sources and sinks of OCS in this region, but rather again acknowledge that atmospheric mixing affects OCS concentrations. At the same time, it seems like a great effort was made to calculate nighttime fluxes with Rn, with no further use of the flux estimates.

*Answer*: the authors agree that referring to sources and sinks in Section 3.1 would be an overstatement. The title has been modified as "*Observed deviations from seasonal cycles by wind directions during stationary measurements*". Section 3.1 and Section 3.2 report the results that contextualized the measurements and the model applications described in the following paragraphs. However, the authors recognize that Section 3.1 and the consequent discussion in Section 4.2 are not well contextualized within this study. They were therefore moved to the supplementary material as Section S1 and Section S1.1, respectively. New numbers were assigned to the remaining Sections in the main text, following the new structure. With regard to Rn measurements, they were performed to identify soil emissions and, consequently, the nighttime COS and $CS_2$ fluxes described in Section 3.1 (previously Section 3.2). The following sentence was introduced in Section S1.1:

"In general, we find depletions of COS only coming from inland, which is likely driven by terrestrial vegetation and soil. *This last, in particular, was measured to be a COS sink during nighttime, as reported in Section 3.1.*"

Further applications of these findings within this study were rather limited. SiB4 data for Lutjewad were requested only for January and February 2018. The average COS nighttime flux was estimated at Lutjewad coordinates over these two months and resulted to be -2.1 ± 0.2 pmol/($m^2$*s). On page 10, Lines 22-24, the following sentence was added:

"*The average SiB4 COS nighttime (9PM – 6AM) flux was retrieved for Lutjewad (53.4°N, 6.3°E) for January and February 2018 and was estimated to be -2.1 ± 0.2 pmol $m^{-2}$ $s^{-1}$*." However, in spite of the limited application presented with the current data, the authors believe these results could provide valuable knowledge for further analyses and/or flux modelling in future studies.

This study misses some context. For example, there are other places in Europe collecting OCS concentration data and an extensive N American dataset that could be used to figure out the seasonal cycle. Some recent efforts to better quantify anthropogenic sources by Sauveur Belviso, who I see has already reviewed this manuscript, would be prudent to include in the interpretation.

In short, this project moves us towards answering several interesting questions in our community, but the analysis is incomplete.

*Answer*: this remark is consistent with the comments of the other referee (Sauveur Belviso) for this study. The authors agree that a further contextualization was needed for this case study, in particular concerning other measurement sites in Europe. The following text has been added to the manuscript:

*Page 3, Lines 21-28:* *Tropospheric COS molar fraction is only monitored in a few sites in Europe. Among these, four monitoring sites are located in Western Europe, within 48°N and 53°N: Mace Head, Ireland (Montzka et al., 2007), Gif-sur-Yvette and Trainou, France (Belviso et al., 2022) and Lutjewad, the Netherlands (Kooijmans et al., 2016). Moreover,*

*COS has been recently monitored discontinuously in Utrecht, the Netherlands (Baartman et al., 2022). Comparing these observations show higher autumn and winter COS molar fraction in the Netherlands than in the comparable sites listed above. This calls for a more thorough investigation of possible local sources in the Netherlands at a local and regional scale.*

*Page 20, Lines 19-23: This approach, combining COS stationary measurements, mobile measurements and models, could be applied in other existing measurement locations. It could allow a broader assessment of local anthropogenic influences, to prevent biases in COS budget and seasonality estimates.*

Minor Comments

Figure 1 and site description: the site description gives context to the Lutjewad tower that is lacking in the map. Maps are difficult to make well and I found myself sketching a separate map to understand the greater context. It would be useful to mention that the ocean, aluminum smelting, wetlands, and winter wheat are all known sources of atmospheric OCS. Figure 1 and several other figures need a more robust caption.

**Answer**: the authors agree that the map could have been drawn in a more informative way. Figure 1 and its caption were modified as follows:

[Figure]

*Figure 2: location of Lutjewad and of the sampling locations in the province of Groningen (NL). The map reports also the major features of the sampling locations and their surrounding areas. Only the locations where emissions were detected will be described in the text.*

Table 1: Is ploughing a source of OCS? Or is the ploughed soil?

**Answer**: *currently, it is believed that the source could be identified in outgassing from the ploughed soil. However, the emissions in this case could not be distinguished between ploughing activites (e.g. agricultural vehicles, fertilization) and ploughed soil. The "source type" description was therefore modified to* Ploughing, soil.

P6, L18: Mentioning why these extra cylinders were collected would be helpful here, even if the details are included in the supplement.

**Answer**: *it is unclear to the authors if the referee was referring to the standard cylinders or to the sampled flasks. Regarding the cylinders, the authors believe that the whole measurement technique and its relative calibration procedure, described in Kooijmans et al. (2016), have been summarized thoroughly in Section 2.2.1. To make it clearer to the reader, the paragraph at page 6, lines 11-18 was modified as follows:*

*"Field standard cylinders are calibrated against NOAA standards in the laboratory before and after each measurement period, to test for drift in molar fraction of gas species. The COS mole fraction measurements of nine cylinders are available, and five cylinders changed less than 2.5 ppt/year, two cylinders decreased by ~10 ppt/year and 2 cylinders decreased by ~30 ppt/year. The four cylinders that drifted more than 10 ppt/year were not used as reference cylinders in the data processing. All of the cylinders were uncoated aluminum cylinders, which, according to experience at NOAA, are more prone to COS mole fractions drift than Aculife treated aluminum cylinders."*

*Regarding the flasks, the paragraph at page 6, line 20 was modified as follows:*

*"To investigate COS seasonal cycle amplitude in Lutjewad, besides the in-situ measurements, we also measured flasks that were sampled at 60 m..."*

P7, L17: Emission rather than exhalation? Or is this a term specific to Rn?

**Answer**: *exhalation is indeed a specific term for Rn.*

P7, L23-24: Is simply taking the average the "done" thing for dealing with Rn emission variability? Can you cite another group or two who have done this and perhaps did a sensitivity analysis or similar?

**Answer**: *this is also a specific method for Rn exhalations estimates (Alhamdi & Abdullah, 2021; Levin et al., 2021; Thabayneh, 2018). In particular, Levin et al. (2021) focuses on*

*radon-tracer method applications and limitations and, while stressing the advantages of high-frequency measurements and a day-to-day variability oscillating between ±10% and ±30%, employs monthly averages in their analyses.*

P7, L26-27: The methods are cited, however, can you give a 1 sentence explanation for why the method only works at night? It seems earlier in the paragraph there is a comparison between daytime and nighttime PBL. While we'd expect photosynthesis to cease at night, making the nighttime fluxes easier. Is that's what's happening here? I know you cite the papers that include more detail on the methods and I could read those and piece it together myself, though as it stands the paragraph here is confusing.

***Answer***: *more than a photosynthesis issue, it is a convection issue: given that the $^{222}$Rn-tracer method is based on vertical gradients in stable (non-convective) conditions, in presence of (vertical) turbulence the method becomes difficult to apply.*

*An example is given for clarification, citing van der Laan et al. (2010): "Another source of uncertainty is the fact that the $^{222}$Rn flux method is based on (vertical) atmospheric gradients which are observed mostly in the evenings and nights when the atmosphere is in general more stable (Fig. 10). Our method is therefore less suitable for estimating surface emissions in the afternoon when vertical mixing is more pronounced. Most of the day is, however, well covered and also the traffic peaks in the mornings and evenings are generally included in our data set. Figure 10 shows furthermore that there is no significant correlation between the height of the flux and the time of the events, which is probably because each event represents a single integrated value that usually includes emissions over several hours during day and night.".*

[Figure]

*Fig. 10.* Distribution of the selected events during the day. Fewer events are observed during 10:00–16:00 when vertical mixing is more pronounced.

P8, L13: What does it mean for a footprint to be negligible?

*Answer: typically, footprint values decrease with the timesteps taken from the start of the simulation. An example is provided in Figure C. This trend changes for every simulation, depending on the parameters leading the model. In the example provided, which refers to 15/02/2018, 9:00 AM, the sum of footprint values starts at 0.57 for the first timestep back in time and decreases to ~0.001 at timestep 38 (4 days and 6 hours back in time), after which it remains stably at 0. For other simulations, these conditions were reached after 8 to 9 days (64 to 72 timesteps). For clarity, page 8, lines 23-26, were modified as follows: "In this analysis, footprints are reliably negligible (their sum over the selected domain being at least 3 orders of magnitude lower than the beginning of the simulation) after 8 to 9 days. Therefore, the simulation timespan is set on 10 days to confidently cover all the potentially significant footprint values".*

[Figure]

Change of sum of footprint values over simulated days back in time

*Figure C: sum of footprint values over timesteps from the start of the simulation (3-hours timesteps for 10 days, for a total of 80 days).*

P16, L25: For a guassian distribution to be a useful model here, certain assumptions must be met. A justification of these would be useful here.

*Answer: the authors acknowledge that major simplifications were made to apply the Gaussian dispersion model for these estimations. The peaks measured during the mobile sampling campaign closely followed a Gaussian shape. Unfortunately, there were no 3D-sonic measurements coupled with this sampling campaign. Furthermore, in most cases the exact source of the emissions remained unknown (e.g. it was not possible to see any industrial chimney, or there was no visible plume). Therefore, it was necessary to assume that emissions occurred at the sampling height. The emissions were reconstructed using the parametrization of $\sigma_y$ and $\sigma_z$ defined after Pasquill-Gifford stability classes (Csanady, 1973), estimated after a Monte Carlo simulation based on distance from the source and wind speed (obtained from approximated estimates from Google Maps and weather data).*

*This, in fact, results in fluxes uncertainties that range between 44% and 92% of the estimated flux means. The authors would like to stress that the Gaussian plume modelling was applied to obtain some rough estimate; the used approach is not considered a reliable representation of reality, but rather a tool to get the best estimate possible given the available data.*

P17, L19 and on: this goes into discussion rather than results.

**Answer***: the authors recognize that parts of this paragraph may sound like a discussion. The paragraph at page 18, lines 16-24*

*"However, it is good to mention that the model resolution might have not been high enough to reproduce the dispersion of emissions in such a limited zone. Moreover, it is possible that other sources could be present nearby Lutjewad, or in general in the areas influencing the observations at the tower. Furthermore, the vertical mixing parameter of the model may have been too fast to correctly simulate the plume transport in such a limited area with stable night conditions. Also, possible indirect emissions of CS2 were not considered in this simulation. In other words, a model with a higher resolution and/or a more detailed database would probably produce a different and more accurate estimate for the missing source in the area. Therefore, the number stated above should be considered as a rough estimate."*

*has been adapted and moved to page 19, lines 15-24.*

P19, L20: The word "prove" is too strong here.

**Answer***: the sentence was modified: "The results presented in Section 3.4 demonstrate the presence of local sources of COS in the province of Groningen."*

P19, L20-30: Too much faith is being put into the STILT analysis. Note that the model is run with imperfect data, the PBL height is often off and this effects the size of the "box".

**Answer***: please see the answer to the following point.*

P20, L29-31: I'm not sure this conclusion is justified.

**Answer***: the authors thank the referee for these valuable remarks and would like to underline how the purpose of this study was not to quantify missing sources with a complete inversion analysis. Rather, the intention of this STILT application was to gain a (qualitative) description of regions that can be influential for COS measurements in Lutjewad. However, while recognizing this model's application limits and flaws, it is interesting to notice how the estimated influences of some known sources are still well in line with the measurements, when combined to STILT simulations. This is particularly noticeable for $CO_2$ (see, for instance, the peaks in Period 1, 2 and 4 in Figure 5 in the main*

*text and the relative linear regressions of modelled results vs measurements in Figure S3 in the supplement). On the other hand, while applying the same model on other periods in time (Period 3) or other species (COS), the same method produces clearly less accurate results. Within this framework, the authors found it reasonable to conclude that in this case the cause for mismatch could be found in missing sources – or peculiar events – rather than in the model's limitations.*

P20, the rest of section 4.2: This reads like speculation when you have an analysis with associated uncertainty to rely on.

**Answer**: *the authors acknowledge a speculative aspect in Section 4.2, which was meant to cover some qualitative aspects of the results presented in Section 3.1. As stated earlier in this response, these sections have been moved to the supplementary material together as Section S1 (previously, Section 3.1) and Section 1.1 (previously, Section 4.2).*

P21, L26: In conclusion, this inversion analysis is incomplete.

**Answer**: *the authors agree this study does not present a thorough inversion analysis. As mentioned previously, this study was not aiming to test the STILT model validity or to estimate missing sources with an inversion approach, but rather to present a case-study application of STILT with known and newly-discovered COS sources. Given the lack of weather data and a rather restricted coverage of the emission ranges in the identified emitters, it was not possible to infer a parametrized emission value for the local sources. However, this approach could be extended to other measurement stations and to more detailed model analysis to provide a proof of concept for the identification of possibly missing sources or sinks surroundings stationary measurements.*

Thank you for your efforts so far. This is an interesting dataset and is moving towards the most interesting application of atmospheric OCS observations.

Mary Whelan

**References used in the responses (excl. web links in the text):**

Alhamdi, W. A., & Abdullah, K. M. S. (2021). Determination of Radium and Radon Exhalation Rate as a Function of Soil Depth of Duhok Province—Iraq. *Journal of Radiation Research and Applied Sciences*, *14*(1), 486–494. https://doi.org/10.1080/16878507.2021.1999719

Levin, I., Karstens, U., Hammer, S., DellaColetta, J., Maier, F., & Gachkivskyi, M. (2021). Limitations of the radon tracer method (RTM) to estimate regional

greenhouse gas (GHG) emissions – a case study for methane in Heidelberg.

*Atmospheric Chemistry and Physics*, *21*(23), 17907–17926.

https://doi.org/10.5194/acp-21-17907-2021

Thabayneh, K. M. (2018). Determination of radon exhalation rates in soil samples

using sealed can technique and CR-39 detectors. *Journal of Environmental*

*Health Science and Engineering*, *16*(2), 121–128.

https://doi.org/10.1007/s40201-018-0298-2

van der Laan, S., Karstens, U., Neubert, R. E. M., Laan-Luijkx, V. D., & Meijer, H. A. J.

(2010). Observation-based estimates of fossil fuel-derived $CO_2$ emissions in

the Netherlands using Δ14C, CO and [222] Radon. *Tellus B: Chemical and Physical*

*Meteorology*, *62*(5), 389–402. https://doi.org/10.1111/j.1600-

0889.2010.00493.x

---

## Referee Report (RR1)

Review of MS egusphere-2023-208R

This new version is acceptable for final publication in BG provided that the following corrections are made, none of them being of major importance.

Page 3 – line 26: "The observations in these studies show higher autumn and winter COS molar fractions in the Netherlands than those at Gif-sur-Yvette and Trainou"… and than those at Mace Head too! It is worth mentioning that autumn and winter COS molar fractions in the Netherlands are higher than in the other European countries (or regions) where COS is monitored in the lower atmosphere.

Page 9 – Figure 2d and page 12 – lines 10-13: Panel 2d identifies the sources influencing Lutjewad in the Ruhr area only during period 4 (see Fig. 4). However, COS enhancements during periods 1, 2 and the last of period 3 were attributed to industry in the Antwerp-Rotterdam region. I think than an illustration targeting the last episode of period 3 should be provided in the main text or in the supplements.

Page 14 – Figure 6: The periods of interest highlighted in yellow in Fig. 4 should be displayed in Fig. 6 too. Please explain why the dashed red curves in Fig. 4 and the upper right panel in Fig. 6 don't look the same.

[Figure]

[Figure]

The recent efforts to better quantify anthropogenic sources of COS in Europe by Belviso et al. (2023) are not mentioned.

References

Belviso S, Pison I, Petit JE, Berchet A, Remaud M, Simon L, Ramonet M, Delmotte M, Kazan V, Yver-Kwok C, and Lopez M. (2023) The Z-2018 emissions inventory of COS in Europe: A semiquantitative multi-data-streams evaluation. Atmos. Environ. 300. https://doi.org/10.1016/j.atmosenv.2023.119689

---

## Author Response (AR2)

Dear Editor, Dear Referees,

First of all, also on behalf of the other co-authors, we would like to thank you for your positive feedbacks and for the opportunity of reviewing this manuscript towards a publication on the Journal of Biogeosciences. Following the last comments received from the Referee, the suggested minor reviews have been accepted and implemented in the manuscript. In particular:

- Page 3, lines 27-32 has been modified as follow: "The observations in these studies show higher autumn and winter COS molar fractions in the Netherlands than those *at Mace Head, Ireland* and at Gif-sur-Yvette and Trainou, France. This calls for a more thorough investigation of possible local sources in the Netherlands at a local and regional scale. *A proper assessment of local sources is also necessary to evaluate the performance of existing databases, such as the one realized by Zumkehr et al. (2018). A recent effort has been reported by Belviso et al. (2023) at a sub-regional level in France.*"
- A new figure (see Figure A below) has been added to the supplementary materials, to exemplify footprints related to the Antwerp-Rotterdam region influence. It is mentioned in the main text on Page 12, lines 25-28: "This period is related to a mixed southern and eastern footprint, which ascribes this share of enhancements to the Antwerp-Rotterdam area and to paper production locations in northern Germany *(an example is shown in Figure S6 in the supplementary material)*."

[Figure]

*Figure A: example of (a) footprint and (b) related COS enhancements for February 9$^{th}$, 2018 (Period 3), retrieved as described in Sect. 2.4 in the main text (see also Figure 2). In this period, part of the measured COS enhancements could be attributed to the Antwerp-Rotterdam region. This is Figure S6 in the supplementary material.*

- Figure 6 has been modified as suggested, including a shading for the relevant periods similarly to Figure 4 (see Figure B below). The discrepancy between the two figures was due to an obsolete scenario being plotted in Figure 6. We would like to thank the Referee once more for noticing this, since it had been overlooked till now. The new Figure 6 is now correctly matching Figure 4.
- The efforts to quantify European anthropogenic COS sources reported in Belviso et al. (2023) have been mentioned in the main text and added to the references (see above).

[Figure]

*Figure B (updated version of Figure 6 in the main text).*

With these, we believe to have addressed all the remarks received so far thoroughly and we hope this manuscript will now be eligible for publication in the Journal of Biogeosciences.

With kind regards, also on behalf of the coauthors,
Alessandro Zanchetta and Huilin Chen

---

## Author Response (AR3)

Dear Editor,

On behalf of the co-authors, we would like to thank you for coordinating this reviewing process and finally accepting this manuscript for final publication in the Journal of Biogeosciences.

We are pleased to upload the necessary materials, together with the following remarks:

1. The literature has been now compiled according to the Copernicus guideline, using the dedicated CSL style as presented in the Copernicus guidelines for publication. We would like to apologize for overlooking this previously during the submission process.
2. The modelled data has been uploaded and is openly accessible at the DOI https://doi.org/10.5281/zenodo.7409361, as reported in the manuscript.

We believe to have covered all the necessary aspects required for publication. Of course, we will remain available and reachable if any further step will be needed.

With kind regards, also on behalf of the coauthors,
Alessandro Zanchetta and Huilin Chen